# Bifurcation and optimal control analysis of HIV/AIDS and COVID-19 co-infection model with numerical simulation

**Belela Samuel Kotola** [1,2]*, **Shewafera Wondimagegnhu Teklu** [2], **Yohannes Fissha Abebaw** [2]

**1** Oda Bultum University, Chiro, Ethiopia, **2** Department of Mathematics, Natural Science, Debre Berhan University, Debre Berhan, Ethiopia

* belelasamuel@gmail.com, belelasmuel@dbu.edu.et

**Data Availability Statement:** All relevant data are within the paper.

**Funding:** The author(s) received no specific funding for this work.

## Abstract

HIV/AIDS and COVID-19 co-infection is a common global health and socio-economic problem. In this paper, a mathematical model for the transmission dynamics of HIV/AIDS and COVID-19 co-infection that incorporates protection and treatment for the infected (and infectious) groups is formulated and analyzed. Firstly, we proved the non-negativity and boundedness of the co-infection model solutions, analyzed the single infection models steady states, calculated the basic reproduction numbers using next generation matrix approach and then investigated the existence and local stabilities of equilibriums using Routh-Hurwiz stability criteria. Then using the Center Manifold criteria to investigate the proposed model exhibited the phenomenon of backward bifurcation whenever its effective reproduction number is less than unity. Secondly, we incorporate time dependent optimal control strategies, using Pontryagin's Maximum Principle to derive necessary conditions for the optimal control of the disease. Finally, we carried out numerical simulations for both the deterministic model and the model incorporating optimal controls and we found the results that the model solutions are converging to the model endemic equilibrium point whenever the model effective reproduction number is greater than unity, and also from numerical simulations of the optimal control problem applying the combinations of all the possible protection and treatment strategies together is the most effective strategy to drastically minimizing the transmission of the HIV/AIDS and COVID-19 co-infection in the community under consideration of the study.

## 1. Introduction

Infectious diseases are diagnostically proven illnesses caused by tiny microorganisms such as viruses, bacteria, fungi, and parasites and have been the leading causes of death throughout the world, for example; viruses cause both COVID-19 and HIV/AIDS infections [1–3].

Human immunodeficiency virus (HIV) is one of the most dangerous viruses that is spreading around the world. AIDS, or acquired immunodeficiency syndrome, is one of the most

**Competing interests:** The authors have declared that no competing interests exist.

devastating epidemics in history, caused by HIV, which has been a worldwide epidemic since 1981 [4–10]. It remains a significant world health issue that impacts almost seventy million people worldwide and has been a significant cause of morbidity and mortality [11,12]. HIV is transmissible through sexual contact, needle sharing, and direct contact with virus-infected blood or other body fluids, as well as from mother to child during giving birth [10,13–15].

In early December 2019, a coronavirus called COVID-19 was reported in Wuhan, China, with symptoms similar to pneumonia. According to reports, it is one of the most devastating infectious diseases caused by the novel coronavirus SARS-CoV-2, which has been a significant impact on the health, social, and economic integration of communities worldwide [16–29]. On March 11, 2020, the World Health Organization (WHO) confirmed it as a global pandemic, and on July 25, 2020, the world total number of COVID-19 infected individuals was 15,762,007, with 640,276 deaths [25,28,29]. It was suspected to be pneumonia or a common cold-like illness, with symptoms such as fatigue, alter in taste, fever, muscular pains, shortness of breath, ironical cough, and sore throat [25,27,30]. Despite massive efforts to reduce the virus's transmission and survivability, the death rate from COVID-19 remains high [15]. COVID-19 can be transmitted through sneezing or coughing droplets expelled from the human lungs, as well as when humans come into contact with contaminated dispatched materials [17,26,31]. Among the unfortunate aspects of the COVID-19 pandemic is that patients over the age of 60 are more likely to be infected than anyone below the age of 60 [31]. It is an extremely infectious contagious agent that has spread throughout most of the world's nations and has a significant impact on the global economy and public health [24,32]. COVID-19 infection may be more common in people with compromised immunity from other infections such as tuberculosis, HIV, pneumonia, and cholera [1,25,33–37]. WHO unanimously implemented vaccination, quarantine, wearing face masks, hand washing with alcohol, and significant discrepancies as possible prevention and control strategies [26,27,31]. Symptomless and pre-symptomatic transmission, a low incidence or lack of dominant systemic symptoms such as fever, airborne transmission that may require a high infectious dose and super-spread events are the essential aspects of COVID-19 spreading that make it challenging to handle [16].

A co-infection is the infection of a single individual with two or more different pathogens or different strains of the same pathogens, leading to co-existence of strains (pathogens) at population level [10]. Co-infection of two or more diseases in one individual is a regular occurrence in today's society [2,14]. Different researchers have investigated that COVID-19 infection could be high in people living with other infections like TB, HIV, and cholera who have compromised immunity [1,8,21,25,30,33–44].

Mathematical modelling approaches have been crucial to provide basic frameworks in order to understand the transmission dynamics of infectious diseases [37]. Many scholars throughout the world have been formulated and analyzed mathematical models to investigate the transmission dynamics of different infectious diseases using ordinary differential equations approach like [2,9,15,17,19,22,23,26–29,31,32,45–47] using stochastic approach like [48], and using fractional order derivative approach like [1,5,49,50]. In the structure of this study, we have reviewed research papers that have been done on the transmission dynamics of different infectious diseases especially co-infections of HIV/AIDS and other infectious diseases. Teklu and Rao [14] constructed and examined HIV/AIDS and pneumonia co-infection model with control measures such as pneumonia vaccination and treatments of pneumonia and HIV/AIDS infections. Hezam et al. [40], formulated a mathematical model for cholera and COVID-19 co-infection which describes the transmission dynamics of COVID-19 and cholera in Yemen. The model analysis examined four controlling measures such as social distancing, lockdown, the number of test kits to control the COVID-19 outbreak, and the number of

susceptible individuals who can get CWTs for water purification. Anwar et al. [15], constructed a mathematical model on COVID-19 with the isolation controlling measure on the COVID-19 infected individuals throughout the community. Ahmed et al. [1] formulated and analyzed HIV and COVID-19 co-infection model with ABC-fractional operator approach to investigate an epidemic prediction of a combined HIV-COVID-19 co-infection model. Numerical simulations were carried out to justify that the disease will stabilize at a later stage when enough protection strategies are taken. Teklu and Terefe [3] analyze COVID-19 and syphilis co-dynamics model to investigate the impacts of intervention measures on the disease transmission.

Similarly, various Scholars have formulated and analyzed mathematical models with optimal control strategies to investigate the effect of prevention and control measures on HIV/AIDS, COVID-19, HIV/AIDS and COVID-19 co-infection and other various infectious diseases transmission throughout nations in the world. For instance, Tchoumi et al. [37] proposed and investigated the co-dynamics of malaria and COVID-19 co-dynamics: with optimal control strategies. The numerical simulation results verifies the theoretical optimal control analysis and illustrates that using malaria and COVID-19 protection measures concurrently can help mitigate there transmission compared with applying single infections protection measures. Omame et al. [25] investigated a mathematical model for the dynamics of COVID-19 infection in order to assess the impacts of prior comorbidity on COVID-19 complications and COVID-19 reinfection with optimal control strategies. The authors recommended that the strategy that prevents COVID-19 infection by comorbid susceptible is the best cost-effective of all the other control strategies for the prevention of COVID-19. Ringa et al. [43] formulated and analyzed a mathematical model on HIV and COVID-19 co-infection with optimal control strategies. Their analysis suggested that COVID19 only prevention strategy is the most effective strategy and it averted about 10,500 new co-infection cases. Keno et al. [51] investigated an optimal control and cost effectiveness analysis of SIRS malaria disease model with temperature variability facto. Their result suggested that the combination of treatment of infected humans and insecticide spraying was proved to be the best efficient and least costly strategy to eradicate the disease. Keno et al. [52] investigated a mathematical model with optimal control strategies for malaria transmission with role of climate variability. Their result suggested that the combination of treated bed net and treatment is the most optimal and least-cost strategy to minimize the malaria. Goudiaby et al. [39] formulated and analyzed a COVID-19 and tuberculosis co-dynamics model with optimal control strategies. They suggested that COVID-19 prevention, treatment and control of co-infection yields a better outcome in terms of the number of COVID-19 cases prevented at a lower percentage of the total cost of this strategy. Asamoah et al. [53] constructed a mathematical model on COVID-19 to investigate optimal control strategies and comprehensive cost-effectiveness. Okosun et al. [54] formulated a mathematical model on HIV/AIDS to investigate the impact of optimal control on the treatment of HIV/AIDS and screening of unaware invectives. Their analysis recommended that the combination of all the control strategies is the most cost-effective strategy. Furthermore, notice that optimal control modeling and cost-effectiveness analysis model have been applied in recent infectious diseases models like [55,56].

As we observed from review of literatures done by various epidemiology and medical scholars, HIV/AIDS and COVID-19 co-infection is a public health concern especially in developing nations of the world. The main purpose of this paper is to investigate the impacts of COVID-19 protection with quarantine, COVID-19 treatment, HIV protection and HIV treatment prevention and controlling strategies on the transmission dynamics of HIV/AIDS and COVID-19 co-infection in the community with mathematical modelling approach. We have reviewed literatures [1,43] invested much effort in studying HIV/AIDS and COVID-19 co-infection, but

did not considered COVID-19 protection with quarantine, COVID-19 treatment, HIV/AIDS protection, and HIV/AIDS treatment as prevention and control strategies simultaneously in a single model formulation which motivates us to undertake this study and fill the gap.

## 2. Mathematical model construction

### 2.1. Basic frameworks of the model

In this paper, we partitioned the total human population at a given time t denoted by $N(t)$, into eleven mutually-exclusive classes depending on their infection status: susceptible class to both COVID-19 and HIV $S(t)$), COVID-19 protection by quarantine class ($C_q(t)$), HIV protected (such as by using condom, limit sexual partners, creating awareness etc.) class ($H_p(t)$), COVID-19 protection by vaccination class ($C_v(t)$), COVID-19 mono-infection class ($C_i(t)$), HIV unaware mono-infection class ($H_u(t)$), HIV aware mono-infection class ($H_a(t)$), HIV unaware and COVID-19 co-infection class ($M_u(t)$), HIV aware and COVID-19 co-infection class ($M_a(t)$), COVID-19 recovery class ($R(t)$), and HIV aware treatment class ($H_t(t)$) so that;

$$N(t) = S(t) + C_q(t) + H_p(t) + C_v(t) + C_i(t) + H_u(t) + H_a(t) + M_u(t) + M_a(t) + C_t(t) + R(t).$$

Since HIV is a chronic infectious disease the susceptible individuals acquires HIV infection at the standard incidence rate given by

$$\lambda_H(t) = \frac{\beta_1}{N}(H_u(t) + \rho_1 H_a(t) + \rho_2 M_u(t) + \rho_3 M_a(t)) \tag{1}$$

where $\rho_3 \geq \rho_2 \geq \rho_1 \geq 1$ are the modification parameters that increase infectivity and $\beta_1$ is the HIV transmission rate. Since COVID-19 is a very acute infection the susceptible individuals acquires COVID-19 infection at the mass action incidence rate as stated in [50,51,54].

$$\lambda_C(t) = \beta_2(C_i(t) + \omega_1 M_u(t) + \omega_2 M_a(t)) \tag{2}$$

where $\omega_2 > \omega_1 > 1$ are the modification parameters that increase infectivity and $\beta_2$ is the COVID-19 transmission rate.

Additional model assumptions

- $k_1$, $k_2$, $k_3$, and $k_4$ where $k_4 = 1 - k_1 - k_2 - k_3$ are portions of the number of recruited individuals those are entering to the susceptible class, the COVID-19 protected class, the HIV protected class and the COVID-19 vaccination class respectively.

- The susceptible class is increased by individuals from the COVID-19 vaccinated class in which those individuals who are vaccinated against COVID-19 but did not respond to vaccination with waning rate of $\rho$ and from COVID-19 recovery with treatment class who develop their temporary immunity by the rate $\eta$.

- COVID-19 vaccine is may not be 100% efficient, so vaccinated individuals also have a chance of being infected with portion $\varepsilon$ of the serotype not covered by the vaccine where $0 \leq \varepsilon < 1$.

- $0 < v \leq 1$ is the modification parameter such that COVID-19 infected individual is less susceptible to HIV infection than a susceptible individuals due to morbidity.

- There is screening and testing mechanisms for the previous and current status in each class.

- The human population distribution is homogeneous in each class.

- HIV treated individuals do not transmit infection to others due to awareness.

- Population of human being is variable.

**Table 1. Biological meaning of model parameters.**

| Parameters | Biological definitions |
|---|---|
| $\mu$ | Human natural mortality rate |
| $\Delta$ | Recruitment of new born and immigrants |
| $\alpha_1$ | COVID-19 protection lose rate |
| $\alpha_2$ | HIV protection lose rate |
| $\varepsilon$ | Proportion not covered by the COVID-19 vaccine |
| $\theta$ | Progression rate |
| $\phi_1, \phi_2$ | Modification parameters |
| $d_1$ | COVID-19 death rate |
| $d_2$ | HIV/AIDS death rate for unaware |
| $d_3$ | HIV/AIDS death rate for aware |
| $\kappa$ | The rate at which COVID-19 infected are recovered by treatment |
| $\gamma$ | HIV aware infection treatment rate |
| $\rho$ | COVID-19 vaccination waning rate |
| $\upsilon$ | Modification parameter |
| $\beta_1$ | HIV/AIDS transmission rate |
| $\beta_2$ | COVID-19 transmission rate |
| $k_1$ | Portion of recruitment entered to susceptible |
| $k_2, k_3$ | Portion of recruitment entered to COVID-19 and HIV protections respectively |
| $k_4$ | Portion of recruitment entered to COVID-19 vaccination class |
| $\delta$ | Co-infection progression rate |
| $\theta_1, \theta_2$ | COVID-19 treatment rates |
| $\eta$ | The rate at which recovered individuals loss temporary immunity |
| $d_4, d_5$ | Co-infected death rates |

- There is no dual-infection transmission simultaneously.

- No vertical HIV transmission.

- No permanent immunity for COVID-19 infection.

In this section using parameters given in Table 1, model variables given in Table 2, and the model basic frame work, and assumptions given in (2.1), the schematic diagram for the transmission dynamics of HIV/AIDS and COVID-19 co-infection is given by Fig 1.

**Table 2. Biological definitions of model variables.**

| Variables | Biological Definitions |
|---|---|
| $S$ | Susceptible class to both HIV and COVID-19 infections |
| $C_q$ | Individuals who are protected by quarantine against COVID-19 |
| $H_p$ | Individuals who are protected against HIV infection |
| $C_v$ | COVID-19 vaccinated class |
| $C_i$ | COVID-19 infected class |
| $H_u$ | Individuals mono-infected with HIV and unaware |
| $H_a$ | Individuals mono-infected with HIV and aware |
| $M_u$ | Co-infected individuals unaware of HIV infection |
| $M_a$ | Co-infected individuals aware of HIV infection |
| $R$ | COVID-19 recovered class |
| $H_t$ | HIV/AIDS treated class |

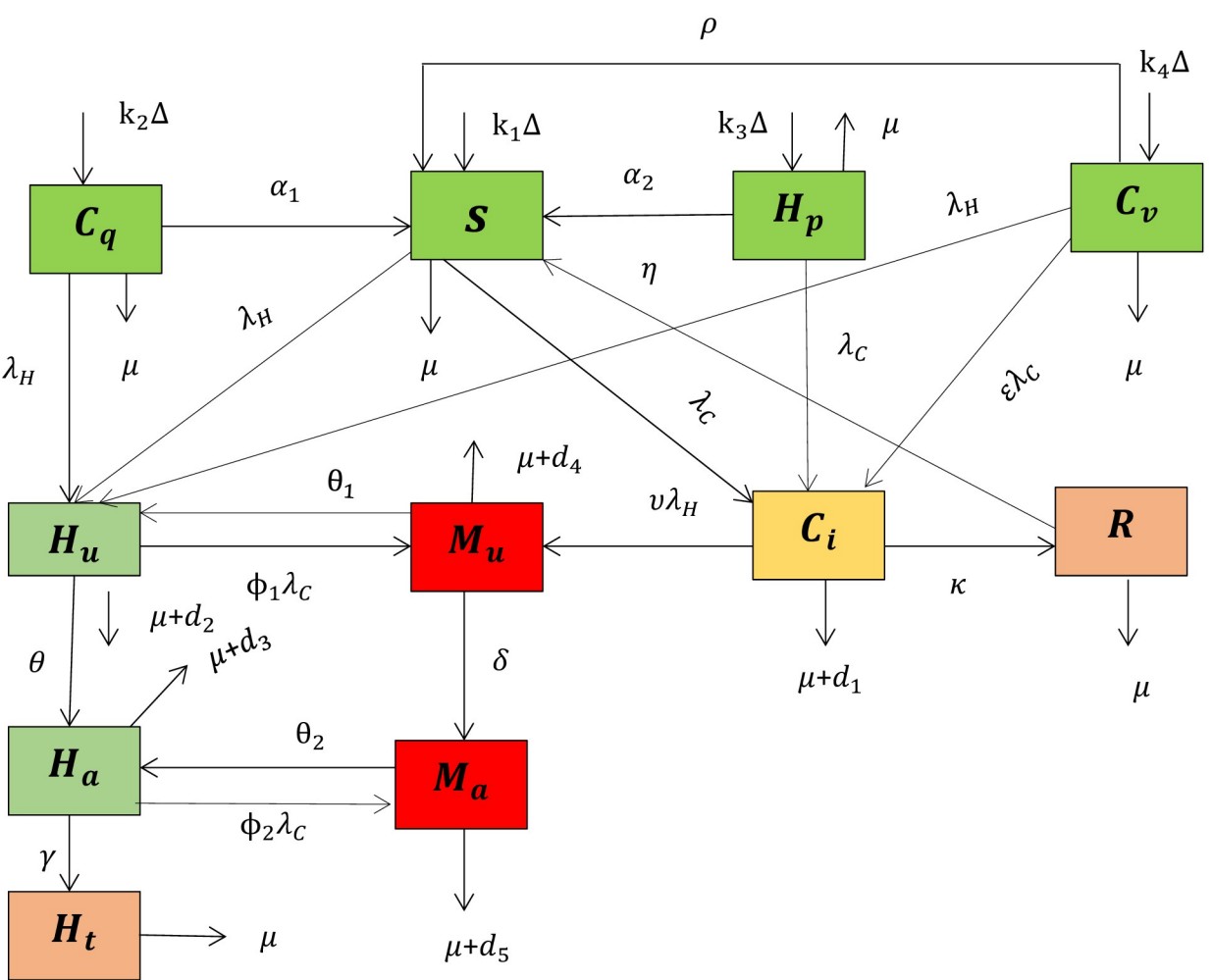

**Fig 1. The schematic diagram of the HIV/AIDS and COVID-19 co-infection transmission dynamics.**

Now using Fig 1 the system of differential equations of the HIV/AIDS and COVID-19 co-infection is given by

$$
\begin{aligned}
\dot{S} &= k_1\Delta + \alpha_1 C_q + \alpha_2 H_p + \rho C_v + \eta R - (\lambda_H + \lambda_C + \mu)S, \\
\dot{C_q} &= k_2\Delta - (\lambda_H + \alpha_1 + \mu)C_q, \\
\dot{H_p} &= k_3\Delta - (\alpha_2 + \mu + \lambda_C)H_p, \\
\dot{C_v} &= k_4\Delta - (\rho + \mu + \lambda_H + \varepsilon\lambda_C)C_v, \\
\dot{C_i} &= \lambda_C S + \lambda_C C_q + \varepsilon\lambda_C C_v - (\mu + d_1 + \kappa + \upsilon\lambda_H)C_i, \\
\dot{H_u} &= \lambda_H S + \lambda_H C_q + \lambda_H C_v + \theta_1 M_u - (\theta + \mu + d_2 + \phi_1\lambda_C)H_u, \\
\dot{H_a} &= \theta H_u + \theta_2 M_a - (\gamma + d_3 + \mu + \phi_2\lambda_C)H_a, \\
\dot{M_u} &= \phi_1\lambda_C H_u + \upsilon\lambda_H C_i - (\mu + d_4 + \delta + \theta_1)M_u, \\
\dot{M_a} &= \delta M_u + \phi_2\lambda_C H_a - (\mu + d_5 + \theta_2)M_a, \\
\dot{R} &= \kappa C_i - (\mu + \eta)R, \\
\dot{H_t} &= \gamma H_a - \mu H_t,
\end{aligned}
\tag{3}
$$

with the corresponding initial conditions

$$S(0) > 0, C_q(0) \geq 0, \ H_p(0) \geq 0, \ C_v(0) \geq 0, C_i(0) \geq 0, H_u(0) \geq 0, H_a(0) \geq 0, M_u(0)$$
$$\geq 0, M_a(0) \geq 0, \ R > 0, \ \text{and } H_t > 0. \quad (4)$$

The sum of all the differential equations in (3) is

$$\dot{N} = \Delta - \mu N - (d_1 C_i + d_2 H_u + d_3 H_a + d_4 M_u + d_5 M_a). \quad (5)$$

## 2.2. The basic qualitative properties of the model (3)

The COVID-19 and HIV/AIDS co-infection model given in Eq (3) is both biologically and mathematically meaningful if and only if all the model solutions (state variables) are non-negative and bounded in the invariant region

$$\Omega = \left\{ \left( S, \ C_q, H_p, C_v, \ C_i, \ H_u, \ H_a, \ M_u, M_a, R, \ H_t \right) \in \mathbb{R}_+^{11}, N \leq \frac{\Lambda}{\mu} \right\} \quad (6)$$

**Theorem 1** (**Positivity of the model solutions**)

Let us given the initial data in Eq (4) then the solutions $S(t)$, $H_p(t)$, $C_v(t)$, $C_i(t)$, $M_u(t)$, $H_u(t)$, $H_a(t)$, $M_a(t)$, $R(t)$, $C_q(t)$, and $H_t(t)$ of the COVID-19 and HIV/AIDS co-infection model (3) are nonnegative for all time $t > 0$.

**Proof**: Let us consider $S(0) > 0$, $C_q(0) > 0$, $H_p(0) > 0$, $C_v(0) > 0$, $C_i(0) > 0$, $H_u(0) > 0$, $H_a(0) > 0$, $M_u(0) > 0$, $M_a(0) > 0$, $R(0) > 0$, and $Ht(0) > 0$ then for all t > 0.

We have to show that $S(t) > 0$, $C_q(t) > 0$, $H_p(t) > 0$, $C_v(t) > 0$, $C_i(t) > 0$, $H_u(t) > 0$, $H_a(t) > 0$, $M_u(t) > 0$, $M_a(t) > 0$, $R(0) > 0$, and $H_t(t) > 0$.

Define: $\tau = \sup\{S(t) > 0, C_q(t) > 0, H_p(t) > 0, C_v(t) > 0, C_i(t) > 0, H_u(t) > 0, H_a(t) > 0, M_u(t) > 0, M_a(t) > 0, R(0) > 0$, and $H_t(t) > 0\}$. Now since the entire co-infection model state variables are positive and all the state variables are continuous, we can justify that $\tau > 0$. If $\tau = +\infty$, then non-negativity holds. But, if $0 < \tau < +\infty$ we will have $S(\tau) = 0$ or $C_q(\tau) = 0$ or $H_p(\tau) = 0$ or $C_v(\tau) = 0$ or $C_i(\tau) = 0$ or $H_u(\tau) =$ or $H_a(\tau) = 0$ or $M_u(\tau) = 0$ or $M_a(\tau) = 0$ or $R(\tau) = 0$ or $H_t(\tau) = 0$.

Here from the first equation of the COVID-19 and HIV/AIDS co-infection model (3) we have got

$$\dot{S} + (\lambda_H + \lambda_C + \mu)S = k_1\Delta + \alpha_1 C_q + \alpha_2 H_p + \rho C_v + \eta R.$$

and integrate using method of integrating factor we have determined the constant value

$$S(\tau) = M_1 S(0) + M_1 \int_0^\tau exp^{\int (\mu + \lambda_H(t) + \lambda_C(t))dt} \left( \alpha_1 C_q + \alpha_2 H_p + \rho C_v + \eta R \right) dt > 0$$

where

$$M_1 = exp^{-\left( \mu\tau + \int_0^\tau (\lambda_H(w) + \lambda_C(w)) \right)} > 0, \ S(0) > 0,$$

and from the meaning of $\tau$, the solutions $C_q(t) > 0$, $H_p(t) > 0$, $C_v(t) > 0$, $R(t) > 0$. Moreover, the exponential function is always positive, then the solution $S(\tau) > 0$ hence $S(\tau) \neq 0$. Thus following the same procedure for $\tau = +\infty$, all the solutions of the COVID-19 and HIV/AIDS co-infection system (3) are non-negative.

**Theorem 2 (The invariant region)**: All the feasible positive solutions of the co-infection model (3) are bounded in the region (6).

**Proof**: Let $\left( S, C_q, H_p, C_v, C_i, H_u, H_a, M_u, M_a, R, H_t \right) \in \mathbb{R}_+^{11}$ is an arbitrary non-negative solution of the system (3) with initial conditions given in Eq (4). Now adding all the differential equations given in Eq (3) we have got the derivative of the total population $N$ which is given in Eq (5) as

$$\dot{N} = \Delta - \mu N - (d_1 C_i + d_2 H_u + d_3 H_a + d_4 M_u + d_5 M_a).$$

Then by ignoring the infections we have determined that $\dot{N} \leq \Delta - \mu N$ and using separation of variables whenever $t \to \infty$, we have obtained that $0 \leq N \leq \frac{\Delta}{\mu}$. Hence, all the positive feasible solutions of the co-infection model (3) entering in to the region given in Eq (6).

**Note**: Since the model (3) solutions are both positive and bounded in the region (6) the HIV/AIDS and COVID-19 co-infection model (3) is both mathematically and biologically meaning full [45,47,57], then we can consider the two mono-infection models, namely; HIV mono-infection and COVID-19 mono-infection models. This is fundamental for the analysis of the COVID-19 and HIV/AIDS co-infection model.

## 3. Analytical result of the models

Before analyzing the HIV/AIDS and COVID-19 co-infection model given in Eq (3), it is very crucial to gain some basic backgrounds about the COVID-19 and HIV/AIDS mono-infection models.

### 3.1. Mathematical analysis of HIV/AIDS mono-infection model

In this subsection we assume there is no COVID-19 infection in the community i.e. $C_q =$, $C_q = C_i = M_u = M_a = R = 0$ in (3) then the HIV/AIDS sub-model is given by

$$\left. \begin{array}{l} \dot{S} = k_1 \Delta + \alpha_2 H_p - (\lambda_H + \mu)S \\ \dot{H}_p = k_3 \Delta - (\alpha_2 + \mu)H_p \\ \dot{H}_u = \lambda_H S - (\theta + \mu + d_2)H_u \\ \dot{H}_a = \theta H_u - (\gamma + d_3 + \mu)H_a \\ \dot{H}_t = \gamma H_u - \mu H_t, \end{array} \right\} \tag{7}$$

where the total population $N_1(t) = S(t) + H_p(t) + H_u(t) + H_a(t) + H_t(t)$, and the HIV sub-model force of infection given by $\lambda_H = \frac{\beta_1}{N_1}(H_u + \rho_1 H_a)$ and initial conditions $S(0) > 0$, $H_p(0) \geq 0$, $H_a(0) \geq 0$, $H_u(0) \geq 0$ and $H_t(0) \geq 0$. In a similar manner of the full co-infection model (3) in the region $\Omega_1 = \left\{ \left( S, H_p, H_u, H_a, H_t \right) \in \mathbb{R}_+^5, N_1 \leq \frac{\Delta}{\mu} \right\}$, it is sufficient to consider the dynamics of the sub-model (7) in $\Omega_1$ as biologically and mathematically well-posed.

### 3.2. Disease-free equilibrium point of HIV mono-infection model (7) local stability

The disease-free equilibrium point of the HIV mono-infection system in (7) is obtained by making its right-hand side is equal to zero and setting the infected classes and treatment class to zero as $H_u = H_a = H_t = 0$ which yields, $S^O = \frac{k_1 \Delta (\alpha_2 + \mu) + \alpha_2 k_3 \Delta}{\mu(\alpha_2 + \mu)}$, $H_p^0 = \frac{k_3 \Delta}{\alpha_2 + \mu}$. Hence the disease-free equilibrium point is given by $E_{HM}^0 = \left( S^0, H_p^0, H_u^0, H_a^0, H_t^0 \right) = \left( \frac{k_1 \Delta (\alpha_2 + \mu) + \alpha_2 k_3 \Delta}{\mu(\alpha_2 + \mu)}, \frac{k_3 \Delta}{\alpha_2 + \mu}, 0, 0, 0 \right)$.

The local stability of the HIV mono-infection model (7) disease-free equilibrium point is examined by its effective reproduction number denoted by $\mathcal{R}_{HM}$, which is calculated by using the next generation operator method determined by Van den Driesch and Warmouth stated in [2]. Applying the method stated in [29], the transmission matrix $F$ and the transition matrix $V$ i.e., for the new infection and the remaining transfer respectively, are given by

$$F = \begin{bmatrix} \dfrac{\beta_1 \theta S^0}{S^O + H_p^0} & \dfrac{\beta_1 \rho_1 \theta S^0}{S^O + H_p^0} & 0 \\ 0 & 0 & 0 \\ 0 & 0 & 0 \end{bmatrix} \text{ and } V = \begin{bmatrix} \theta + \mu + d_2 & 0 & 0 \\ -\theta & \gamma + d_3 + \mu & 0 \\ 0 & -\gamma & \mu \end{bmatrix}.$$

After some computations we have determined that

$$V^{-1} = \begin{bmatrix} \dfrac{1}{(\theta + \mu + d_2)} & 0 & 0 \\ \dfrac{\theta}{(\theta + \mu + d_2)(\gamma + \mu + d_3)} & \dfrac{1}{(\gamma + \mu + d_3)} & 0 \\ \dfrac{\gamma\theta}{\mu(\theta + \mu + d_2)(\gamma + \mu + d_3)} & \dfrac{\gamma}{\mu(\gamma + \mu + d_3)} & \dfrac{1}{\mu} \end{bmatrix},$$

and

$$FV^{-1} = \begin{bmatrix} \dfrac{\beta_1 S^0}{(S^0 + H_p^0)((\theta + \mu + d_2))} + \dfrac{\beta_1 \rho_1 \theta S^0}{\left(S^0 + H_p^0\right)(\theta + \mu + d_2)(\gamma + \mu + d_3)} & \dfrac{\beta_1 \rho_1 \theta S^0}{\left(S^0 + H_p^0\right)(\gamma + \mu + d_3)} & 0 \\ 0 & 0 & 0 \\ 0 & 0 & 0 \end{bmatrix}.$$

Then, the effective reproduction number of the HIV mono-infection model (7) is defined as the largest eigenvalue in magnitude of the next generation matrix, $FV^{-1}$ given by

$$\mathcal{R}_{HM} = \frac{\beta_1(1 - k_3)(\alpha_2 + \mu) + \beta_1\alpha_2 k_3}{(\alpha_2 + \mu)(\theta + \mu + d_2)} + \frac{\beta_1\rho_1\theta(1 - k_3)(\alpha_2 + \mu) + \beta_1\rho_1\theta\alpha_2 k_3}{(\theta + \mu + d_2)(\gamma + \mu + d_3)}.$$

The value $\mathcal{R}_{HM}$ is defined as the total average number of secondary HIV unaware and HIV aware infection cases acquired from a typical HIV unaware or HIV aware individual during his/her effective infectious period in a susceptible population. The threshold result $\mathcal{R}_{HM}$ is the effective reproduction number for HIV mono-infection.

**Theorem 3**: The disease-free equilibrium point of the HIV mono-infection model given in Eq (7) is locally asymptotically stable (LAS) if $\mathcal{R}_{HM} < 1$, and it is unstable if $\mathcal{R}_{HM} > 1$.

**Proof**: The local stability of the disease-free equilibrium point of HIV mono-infection model (7) is evaluated by applying the Routh-Hurwitz stability criteria stated in [52].

The Jacobian matrix of the HIV mono-infection model given in Eq (7) at the disease-free equilibrium point $E_{HM}^0$ is given by

$$J\left(E_{HM}^0\right) = \begin{bmatrix} -\mu & \alpha_2 & -\dfrac{\beta_1 S^0}{S^0 + H_p^0} & -\dfrac{\beta_1 S^0 \rho_1}{S^0 + H_p^0} & 0 \\ 0 & -(\alpha_2 + \mu) & 0 & 0 & 0 \\ 0 & 0 & \dfrac{\beta_1 S^0}{S^0 + H_p^0} - (\theta + \mu + d_2) & \dfrac{\beta_1 S^0 \rho_1}{S^0 + H_p^0} & 0 \\ 0 & 0 & \theta & -(\gamma + d_3 + \mu) & 0 \\ 0 & 0 & 0 & \gamma & -\mu \end{bmatrix}.$$

Then the corresponding characteristic equation of the Jacobian matrix $J(E_{HM}^0)$ is given by

$$\begin{vmatrix} -\mu - \lambda & \alpha_2 & -\dfrac{\beta_1 S^0}{S^0 + H_p^0} & -\dfrac{\beta_1 S^0 \rho_1}{S^0 + H_p^0} & 0 \\ 0 & -(\alpha_2 + \mu) - \lambda & 0 & 0 & 0 \\ 0 & 0 & \dfrac{\beta_1 S^0}{S^0 + H_p^0} - (\theta + \mu + d_2) - \lambda & -\dfrac{\beta_1 S^0}{S^0 + H_p^0} & 0 \\ 0 & 0 & \theta & -(\gamma + d_3 + \mu) - \lambda & 0 \\ 0 & 0 & 0 & \gamma & -\mu - \lambda \end{vmatrix} = 0,$$

$$\Rightarrow (-\mu - \lambda)(-(\alpha_2 + \mu) - \lambda)(-\mu - \lambda)\left[ \left( -\dfrac{\beta_1 S^0}{S^0 + H_p^0} - (\theta + \mu + d_2) - \lambda \right)(-(\gamma + d_3 + \mu) - \lambda) - \dfrac{\theta \beta_1 S^0 \rho_1}{S^0 + H_p^0} \right]$$

$$= 0.$$

Finally we have determined

$$(-\mu - \lambda)(-(\alpha_2 + \mu) - \lambda)(-\mu - \lambda)(\lambda^2 + a\lambda + b) = 0,$$

where $a = (\gamma + d_3 + \mu) + (\theta + \mu + d_2) - \frac{\beta_1 S^0}{S^0 + H_p^0}$, and $b =$

$-\left( \frac{\beta_1 S^0}{S^0 + H_p^0} - (\theta + \mu + d_2) \right)(\gamma + d_3 + \mu) - \frac{\theta \beta_1 S^0 \rho_1}{S^0 + H_p^0} = (\theta + \mu + d_2)(\gamma + d_3 + \mu) -$

$(\gamma + d_3 + \mu)\frac{\beta_1 S^0}{S^0 + H_p^0} - \frac{\theta \beta_1 S^0 \rho_1}{S^0 + H_p^0} = (\theta + \mu + d_2)(\gamma + d_3 + \mu)\left( 1 - \frac{\beta_1 \rho_1 \theta(1 - k_3)(\alpha_2 + \mu) + \beta_1 \rho_1 \theta \alpha_2 k_3}{(\theta + \mu + d_2)(\gamma + \mu + d_3)} \right) =$

$(\theta + \mu + d_2)(\gamma + d_3 + \mu)(1 - \mathcal{R}_{HM})$.

Then we have got $\lambda_1 = -\mu < 0$ or $\lambda_2 = -(\alpha_2 + \mu) < 0$ or $\lambda_3 = -\mu < 0$ or

$$\lambda^2 + a\lambda + b = 0 \tag{8}$$

On Eq (8) we applied Routh-Hurwitz stability criteria stated in [47] and we have determined that both eigenvalues are negative if $\mathcal{R}_{HM} < 1$. Furthermore, we can conclude that the disease-free equilibrium point of the model (7) is locally asymptotically stable whenever $\mathcal{R}_{HM} < 1$ since all the eigenvalues are negative when $\mathcal{R}_{HM} < 1$. The biological meaning of Theorem 3 can be stated as HIV infection can be eradicated from the population (whenever $\mathcal{R}_{HM} < 1$) if the initial size of the sub-populations of the HIV mono-infection model given in Eq (7) is in the basin of attraction of the disease-free equilibrium point $E_{HM}^0$.

### 3.3. Existence of HIV mono-infection endemic equilibrium point(s)

Let $E^*_{HM} = \left( S^*, H^*_p, H^*_u, H^*_a, H^*_t \right)$ be an arbitrary endemic equilibrium point of the HIV mono-infection model (7) which can be determined by making the right hand side of Eq (7) as zero. The after a number of steps of computations we have got

$$S^* = \frac{k_1 \Delta m_1 + \alpha_2 k_3 \Delta}{m_1 \left( \lambda^*_H + \mu \right)}, \ H^*_p = \frac{k_3 \Delta}{m_1}, \ H^*_u = \frac{k_1 \Delta m_1 \lambda^*_H + \alpha_2 k_3 \Delta \lambda^*_H}{m_1 m_2 \left( \lambda^*_H + \mu \right)},$$

$$H^*_a = \frac{k_1 \Delta \theta m_1 \lambda^*_H + \alpha_2 k_3 \Delta \theta \lambda^*_H}{m_1 m_2 m_3 \left( \lambda^*_H + \mu \right)}, \ H^*_t = \frac{k_1 \Delta \theta \gamma m_1 \lambda^*_H + \alpha_2 k_3 \Delta \theta \gamma \lambda^*_H}{\mu m_1 m_2 m_3 \left( \lambda^*_H + \mu \right)}, \quad (9)$$

where $m_1 = (\alpha_2 + \mu)$, $m_2 = (\theta + \mu + d_2)$, and $m_3 = (\gamma + d_3 + \mu)$.

Now substitute $H^*_u$ and $H^*_a$ given in Eq (9) in to the HIV/AIDS force of infection

$$\lambda^*_H = \frac{\beta_1 H^*_u + \rho_1 H^*_a}{S^* + H^*_p + H^*_u + H^*_a + H^*_t}.$$

Then we have the result

$$(m_5 + m_6 \lambda^*_H - m_4) \lambda^*_H = 0. \quad (10)$$

where $m_4 = \beta_1 k_1 \Delta m_1 m_3 m_3 \mu + \beta_1 \alpha_2 k_3 \Delta m_3 m_3 \mu + \beta_1 \rho_1 k_1 \Delta \theta m_1 m_3 \mu + \beta_1 \rho_1 \alpha_2 k_3 \Delta \theta m_3 \mu$, $m_5 = k_1 \Delta m_1 m_2 m_3 \mu + \alpha_2 k_3 \Delta m_2 m_3 \mu + k_3 \Delta m_2 m_3 \mu \mu$, $m_6 = k_3 \Delta m_2 m_3 \mu + k_1 \Delta m_1 m_3 \mu + \alpha_2 k_3 \Delta m_3 \mu + k_1 \Delta \theta m_1 \mu + \alpha_2 k_3 \Delta \theta \mu + k_1 \Delta \theta \gamma m_1 + \alpha_2 k_3 \Delta \theta \gamma$.

Then the non-zero solution of (10) is $\lambda^*_H = \frac{m_4 - m_5}{m_6}$. Therefore, the required non-zero solution (force of infection is obtained as $\lambda^*_H = \frac{[k_1 \Delta m_1 m_2 m_3 \mu + k_3 \Delta m_2 m_3 \mu (\alpha_2 + \mu)](\mathcal{R}_{HM} - 1)}{k_3 \Delta m_2 m_3 \mu + k_1 \Delta m_1 m_3 \mu + \alpha_2 k_3 \Delta m_3 \mu + k_1 \Delta \theta m_1 \mu + \alpha_2 k_3 \Delta \theta \mu + k_1 \Delta \theta \gamma m_1 + \alpha_2 k_3 \Delta \theta \gamma}$. Then we have got $\lambda^*_H > 0$ whenever $\mathcal{R}_{HM} > 1$. Thus, the HIV/AIDS mono-infection model (7) has a unique positive endemic equilibrium point if and only if $\mathcal{R}_{HM} > 1$.

**Theorem 4**: The HIV/AIDS mono-infection model given in (7) has a unique endemic equilibrium point if and only if $\mathcal{R}_{HM} > 1$.

## 3.4. COVID-19 sub-model analysis

The corresponding COVID-19 sub-model of the system (3) is determined by making $H_p = H_a = H_u = M_u = M_a = H_t = 0$, and it is given by

$$\left. \begin{aligned} \dot{S} &= k_1 \Delta + \alpha_1 C_q + \rho C_v + \eta R - (\lambda_C + \mu) S, \\ \dot{C}_q &= k_2 \Delta - (\alpha_1 + \mu) C_q, \\ \dot{C}_v &= k_4 \Delta - (\rho + \mu + \varepsilon \lambda_C) C_v, \\ \dot{C}_i &= \lambda_C S + \varepsilon \lambda_C C_v - (\mu + d_1 + \kappa) C_i, \\ \dot{R} &= \kappa C_i - (\mu + \eta) R, \end{aligned} \right\} \quad (11)$$

with COVID-19 infection initial conditions $S(0) > 0, C_q(0) \geq 0, C_v(0) \geq 0, C_i(0) \geq 0, R(0) \geq 0$, total population $N_2(t) = S(t) + C_q(t) + C_v(t) + C_i(t) + R(t)$, and COVID-19 force of infection given by $\lambda_C = \beta_2 C_i(t)$. Here like the full model (3) and the HIV/AIDS sub-model (7) in the region $\Omega_2 = \left\{ \left( S, C_q, C_v, C_i, R \right) \in \mathbb{R}^5_+, N_2 \leq \frac{\Delta}{\mu} \right\}$, it is sufficient to consider the dynamics of model (11) in $\Omega_2$ be both biologically and mathematically meaningful.

**3.4.1. Local stability of COVID-19 mono-infection model (11) Disease-free equilibrium.** Disease-free equilibrium point of the COVID-19 mono-infection model (11) is

obtained by making its right-hand side as zero and setting the infected class and recovered with treatment class to zero as $C_i = R = 0$ and after some simple steps of calculations we have determined that $S^0 = \frac{k_1\Delta(\alpha_1+\mu)(\rho+\mu)+\alpha_1 k_2\Delta(\rho+\mu)+k_4\Delta\rho(\alpha_1+\mu)}{\mu(\alpha_1+\mu)(\rho+\mu)}$, $C_q^0 = \frac{k_2\Delta}{\alpha_1+\mu}$, and $C_i^0 = \frac{k_4\Delta}{\rho+\mu}$. Hence the COVID-19 mono-infection model (11) disease-free equilibrium point is given by

$$
\begin{aligned}
E_{PM}^0 &= (S^0, C_q^0, C_v^0, C_i^0, R^0) \\
&= \left( \frac{k_1\Delta(\alpha_1+\mu)(\rho+\mu)+\alpha_1 k_2\Delta(\rho+\mu)+k_4\Delta\rho(\alpha_1+\mu)}{\mu(\alpha_1+\mu)(\rho+\mu)}, \frac{k_2\Delta}{\alpha_1+\mu}, \frac{k_4\Delta}{>\rho+\mu}, 0, 0 \right).
\end{aligned}
$$

Here we are applying the Van Den Driesch and Warmouth next-generation matrix approach stated in [2] to determine the COVID-19 mono-infection model (11) effective reproduction number $\mathcal{R}_C$. After long computations, we have determined the transmission matrix given by

$$
F = \begin{bmatrix} \beta_2 S^0 + \varepsilon\beta_2 C_i^0 & 0 \\ 0 & 0 \end{bmatrix},
$$

and the transition matrix given by

$$
V = \begin{bmatrix} \mu + d_1 + \kappa & 0 \\ -\kappa & \mu + \eta \end{bmatrix}.
$$

Then using Mathematica we have determined as

$$
V^{-1} = \begin{bmatrix} \dfrac{1}{\mu + d_1 + \kappa} & 0 \\ \dfrac{\kappa}{(\mu + d_1 + \kappa)(\mu + \eta)} & \dfrac{1}{\mu + \eta} \end{bmatrix} \text{ and } FV^{-1} = \begin{bmatrix} \dfrac{\beta_2 S^0 + \varepsilon\beta_2 C_i^0}{\mu + d_1 + \kappa} & 0 \\ 0 & 0 \end{bmatrix}.
$$

The characteristic equation of the matrix $FV^{-1}$ is $\begin{vmatrix} \dfrac{\beta_2 S^0 + \varepsilon\beta_2 C_i^0}{\mu + d_1 + \kappa} - \lambda & 0 \\ 0 & 0 - \lambda \end{vmatrix} = 0.$

Then the spectral radius (effective reproduction number $\mathcal{R}_C$) of $FV^{-1}$ of the COVID-19 mono-infection model (11) is $\mathcal{R}_C = \frac{\beta_2 S^0 + \varepsilon\beta_2 C_i^0}{\mu + d_1 + \kappa} = \frac{\beta_2 k_1\Delta(\alpha_1+\mu)(\rho+\mu)+\beta_2\alpha_1 k_2\Delta(\rho+\mu)+\beta_2 k_4\Delta\rho(\alpha_1+\mu)+\beta_2\varepsilon k_4\Delta\mu(\alpha_1+\mu)}{\mu(\alpha_1+\mu)(\rho+\mu)(\mu+d_1+\kappa)}$.

**Theorem 5**: The Disease-free equilibrium point $E_{CM}^0$ of the COVID-19 mono-infection model (11) is locally asymptotically stable if $\mathcal{R}_C < 1$ otherwise unstable.

**Proof**: The local stability of the disease-free equilibrium of the system (11) at point $E_{CM}^0 = \left( \frac{k_1\Delta(\alpha_1+\mu)(\rho+\mu)+\alpha_1 k_2\Delta(\rho+\mu)+k_4\Delta\rho(\alpha_1+\mu)}{\mu(\alpha_1+\mu)(\rho+\mu)}, \frac{k_2\Delta}{\alpha_1+\mu}, \frac{k_4\Delta}{\rho+\mu}, 0, 0 \right)$ can be studied from its Jacobian matrix and Routh-Hurwitz stability criteria. The Jacobian matrix of the dynamical system at the disease-

free equilibrium point is given by

$$
J\left(E_{CM}^{0}\right) = \begin{bmatrix}
-\mu & \alpha_1 & \rho & -\beta_2 S^0 & \eta \\
0 & -(\alpha_1 + \mu) & 0 & 0 & 0 \\
0 & 0 & -(\rho + \mu) & -\beta_2 \varepsilon C_i^0 & 0 \\
0 & 0 & 0 & \beta_2 A_1^0 + \beta_2 \varepsilon C_i^0 - (\mu + d_1 + \kappa) & 0 \\
0 & 0 & 0 & \kappa & -(\mu + \eta)
\end{bmatrix}.
$$

Then the characteristic equation of the above Jacobian matrix is given by

$$
\begin{vmatrix}
-\mu & \alpha_1 & \rho & -\beta_2 S^0 & \eta \\
0 & -(\alpha_1 + \mu) & 0 & 0 & 0 \\
0 & 0 & -(\rho + \mu) & -\beta_2 \varepsilon C_i^0 & 0 \\
0 & 0 & 0 & M & 0 \\
0 & 0 & 0 & \kappa & -(\mu + \eta)
\end{vmatrix} = 0,
$$

where $M = \beta_2 S^0 + \beta_2 \varepsilon C_i^0 - (\mu + d_1 + \kappa)$ and after some steps of computations we have got $\lambda_1 = -\mu < 0$ or $\lambda_2 = -(\alpha_1 + \mu) < 0$ or $\lambda_3 = -(\rho + \mu) < 0$ or $\lambda_4 = \beta_2 S^0 + \beta_2 \varepsilon C_i^0 - (\mu + d_1 + \kappa) = (\mu + d_1 + \kappa)\left[\frac{\beta_2 S^0 + \beta_2 \varepsilon C_i^0}{\mu + d_1 + \kappa} - 1\right] = \mu + d_1 + \kappa)[\mathcal{R}_C - 1] < 0$ if $\mathcal{R}_C < 1$ or $\lambda_5 = -(\mu + \eta) < 0$.

Therefore, since all the eigenvalues of the characteristics polynomials of the system (11) are negative if $\mathcal{R}_C < 1$ the disease-free equilibrium point of the COVID-19 mono-infection model (11) is locally asymptotically stable.

**3.4.2. Existence of endemic equilibrium point (s) of the COVID-19 mono-infection model.** Before checking the global stability of the disease-free equilibrium point of the COVID-19 mono-infection model (11), we shall find the possible number of endemic equilibrium point(s) of the model (11). Let $E_C^* = (S^*, C_q^*, C_v^*, C_i^*, R^*)$ be the endemic equilibrium point of COVID-19 mono-infection and $\lambda_C^* = \beta_2 C_i^*$ be the COVID-19 mono-infection mass action incidence rate ("force of infection") at the equilibrium point. To find equilibrium point (s) for which COVID-19 mono-infection is endemic in the population, the equations are solved in terms of $\lambda_C^* = \beta_2 C_i^*$ at an endemic equilibrium point. Now setting the right-hand sides of the equations of the model to zero (at steady state) gives

$$
S^* = \frac{b_5(b_2 + \varepsilon\lambda_C^*)^2 + b_6(b_2 + \varepsilon\lambda_C^*)^2 + b_7(b_2 + \varepsilon\lambda_C^*) + b_8\lambda_C^*}{b_1 b_3 b_4 (b_2 + \varepsilon\lambda_C^*)^2 (\lambda_C^* + \mu) - b_1 \eta\kappa(b_2 + \varepsilon\lambda_C^*)^2 \lambda_C^*}, \quad C_q^* = \frac{k_2 \Delta}{b_1}, \quad C_v^* = \frac{k_4 \Delta}{(b_2 + \varepsilon\lambda_C^*)},
$$

$$
C_i^* = \frac{b_5(b_2 + \varepsilon\lambda_C^*)^2 \lambda_C^* + b_6(b_2 + \varepsilon\lambda_C^*)^2 \lambda_C^* + (b_2 b_7 \lambda_C^* + b_7 \varepsilon\lambda_C^{*2})}{b_{12}(b_2 + \varepsilon\lambda_C^*)^2 (\lambda_C^* + \mu) - b_{13}(b_2 + \varepsilon\lambda_C^*)^2 \lambda_C^*} + \frac{b_8 \lambda_C^{*2} + (b_2 b_9 + b_9 \varepsilon\lambda_C^*)(\lambda_C^{*2} + \mu\lambda_C^*) - b_{10}\lambda_C^{*2} - b_{11}\lambda_C^{*3}}{b_{12}(b_2 + \varepsilon\lambda_C^*)^2 (\lambda_C^* + \mu) - b_{13}(b_2 + \varepsilon\lambda_C^*)^2 \lambda_C^*},
$$

and

$$
R^* = \frac{\kappa B_5^*}{b_4},
$$

where $b_1 = \alpha_1 + \mu$, $b_2 = \rho + \mu$, $b_3 = \mu + d_1 + \kappa$, $b_4 = \mu + \eta$, $b_5 = k_1\Delta b_1 b_3 b_4$, $b_6 = \alpha_1 k_2 \Delta b_3 b_4$, $b_7 =$

$\rho k_4 \Delta b_1 b_3 b_4$, $b_8 = k_4 \Delta b_1 \eta \kappa \varepsilon$, $b_9 = b_1 b_3 b_4 k_4 \Delta \varepsilon$, $b_{10} = b_2 b_1 \eta \kappa k_4 \Delta \varepsilon$, $b_{11} = b_1 k_4 \Delta \varepsilon \eta \kappa \varepsilon$, $b_{12} = b_1 b_3 b_3 b_4$, $b_{13} = b_1 b_3 \eta \kappa$.

Then we have substituted $C_i^* = \frac{b_5 \left(b_2 + \varepsilon \lambda_C^*\right)^2 \lambda_C^* + b_6 \left(b_2 + \varepsilon \lambda_C^*\right)^2 \lambda_C^* + \left(b_2 b_7 \lambda_C^* + b_7 \varepsilon \lambda_C^{*2}\right)}{b_{12}\left(b_2 + \varepsilon \lambda_C^*\right)^2 \left(\lambda_C^* + \mu\right) - b_{13}\left(b_2 + \varepsilon \lambda_C^*\right)^2 \lambda_C^*} +$

$\frac{b_8 \lambda_C^{*2} + \left(b_2 b_9 + b_9 \varepsilon \lambda_C^*\right)\left(\lambda_C^{*2} + \mu \lambda_C^*\right) - b_{10} \lambda_C^{*2} - b_{11} \lambda_C^{*3}}{b_{12}\left(b_2 + \varepsilon \lambda_C^*\right)^2 \left(\lambda_C^* + \mu\right) - b_{13}\left(b_2 + \varepsilon \lambda_C^*\right)^2 \lambda_C^*}$ in the COVID-19 force of infection given by $\lambda_C^* = \beta_2 R^*$ we have got the non-zero solution of $\lambda_C^*$ is obtained from the cubic equation

$$c_3 \lambda_C^{*3} + c_2 \lambda_C^{*2} + c_1 \lambda_C^* + a_0 = 0, \tag{12}$$

where

$$
\begin{aligned}
&c_3 = b_{12}\varepsilon^2 - b_{13}\varepsilon^2 > 0 \\
&c_2 = 2b_2 b_{12}\varepsilon + b_{12}\mu\varepsilon^2 - 2b_2 b_{13}\varepsilon - b_5\varepsilon^2 - b_6\varepsilon^2 - b_9\varepsilon + b_{11}, \\
&c_1 = b_2^2 b_{12} + b_{10} + 2b_2 b_{12}\varepsilon\mu - b_2^2 b_{13} - 2b_2 b_5\varepsilon - 2b_2 b_6\varepsilon - b_7\varepsilon - b_8 - b_2 b_9 - b_9\mu\varepsilon, \\
&c_0 = b_1 b_2 b_3 b_4 [1 - \mathcal{R}_C] > 0 \text{ if } \mathcal{R}_C < 1.
\end{aligned}
\tag{13}
$$

It can be seen from and (13) that $c_3 > 0$ (since the entire model parameters are nonnegative). Furthermore, $c_0 > 0$ whenever $\mathcal{R}_C < 1$. Thus, the number of possible positive real roots the polynomial (12) can have depends on the signs of $c_1$, and $c_2$. This can be analyzed using the Descartes' rule of signs on the cubic $f(x) = c_3 x^3 + c_2 x^2 + c_1 x + c_0$ (with $= x = \lambda_C^*$). Hence, the following results are established.

**Theorem 6**: The COVID-19 mono-infection model (11) could have

(a). a unique endemic equilibrium point if $\mathcal{R}_C > 1$ either of the following holds.

 (i) $c_1 > 0$ and $c_2 > 0$.

 (ii) $c_1 < 0$ and $c_2 < 0$.

(b). more than one endemic equilibrium point if $\mathcal{R}_C > 1$ either of the following holds.

 (i) $c_1 > 0$ and $c_2 < 0$.

 (ii) $c_1 < 0$ and $c_2 > 0$.

(c). two endemic equilibrium points if $\mathcal{R}_C < 1$, $c_1 < 0$ and $c_2 < 0$.

Here, item (c) shows the happening of the backward bifurcation in the model (11) i.e., the locally asymptotically stable disease-free equilibrium point co-exists with a locally asymptotically stable endemic equilibrium point if $\mathcal{R}_C < 1$; examples of the existence of backward bifurcation phenomenon in mathematical epidemiological models, and the causes, can be seen in [8,17,26,31,58–60]. The epidemiological consequence is that the classical epidemiological requirement of having the reproduction number $\mathcal{R}_C$ to be less than one, even though necessary, is not sufficient for the effective control of the disease. The existence of the backward bifurcation phenomenon in sub-model (11) is now explored.

**Theorem 7**: The COVID-19 mono-infection model (11) exhibits backward bifurcation at $\mathcal{R}_C = 1$ whenever the inequality $D_2 > D_1$ holds, where $D_1 = \frac{-\beta_2 \beta^* x_1^0 (\rho + \mu)(\mu + \eta) - \beta_2 \beta^* \varepsilon x_3^0 \rho(\mu + \eta) - \beta_2 \varepsilon \beta^* \varepsilon x_3^0 \mu(\mu + \eta)}{\mu(\rho + \mu)(\mu + \eta)}$ and $D_2 = \frac{\beta_2 \kappa \eta (\rho + \mu)}{\mu(\rho + \mu)(\mu + \eta)}$.

In this section, we have used the center manifold theory stated in [60] to ascertain the local asymptotic stability of the endemic equilibrium due to the convolution of the first approach (eigenvalues of the Jacobian). To make use of the center manifold theory, the following change of variables is made by symbolizing $S = x_1$, $C_p = x_2$, $C_v = x_3$, $C_i = x_4$ and $R = x_5$ such that $N_2 = x_1$

$+ x_2 + x_3 + x_4 + x_5$. Furthermore, by using vector notation $X = (x_1, x_2, x_3, x_4, x_5)^T$, the COVID-19 mono-infection model (11) can be written in the form $\frac{dX}{dt} = F(X)$ with $F = (f_1, f_2, f_3, f_4, f_5)^T$, as follows

$$
\begin{aligned}
\frac{dx_1}{dt} &= f_1 = k_1\Delta + \alpha_1 x_2 + \rho x_3 + \eta x_5 - \mu x_1 - \lambda_C x_1, \\
\frac{dx_2}{dt} &= f_2 = k_2\Delta - (\alpha_1 + \mu)x_2, \\
\frac{dx_3}{dt} &= f_4 = k_4\Delta - (\rho + \mu + \varepsilon\lambda_C)x_3, \\
\frac{dx_4}{dt} &= \lambda_C x_1 + \varepsilon\lambda_C x_3 - (\mu + d_1 + \kappa)x_4, \\
\frac{dx_5}{dt} &= \kappa x_4 - (\mu + \eta)x_5,
\end{aligned}
\tag{14}
$$

with $\lambda_C = \beta_2 x_4$ then the method entails evaluating the Jacobian of the system (14) at the DFE point $E^0_{CM}$, denoted by $J(E^0_{CM})$ and this gives us

$$
J(E^0_{CM}) = \begin{pmatrix}
-\mu & \alpha_2 & \rho & -\beta_2 x_1^0 & \eta \\
0 & -(\alpha_1 + \mu) & 0 & 0 & 0 \\
0 & 0 & -(\rho + \mu) & -\beta_2\varepsilon x_3^0 & 0 \\
0 & 0 & 0 & \beta_2 x_1^0 + \beta_2\varepsilon x_3^0 - (\mu + d_1 + \kappa) & 0 \\
0 & 0 & 0 & \kappa & -(\mu + \eta)
\end{pmatrix}.
$$

Consider, $\mathcal{R}_C = 1$ and suppose that $\beta_2 = \beta^*$ is chosen as a bifurcation parameter. From $\mathcal{R}_C = 1$ as $\mathcal{R}_C = \frac{\beta_2 x_2^0 + \varepsilon\beta_2 x_4^0}{\mu + d_1 + \kappa} = \frac{\beta_2 k_1\Delta(\alpha_1+\mu)(\rho+\mu) + \beta_2\alpha_1 k_2\Delta(\rho+\mu) + \beta_2 k_4\Delta(\alpha_1+\mu)(\rho+\mu\varepsilon)}{\mu(\alpha_1+\mu)(\rho+\mu)(\mu+d_1+\kappa)} = 1$.

Solving for $\beta_2$ we have got $\beta_2 = \beta^* = \frac{\mu(\alpha_1+\mu)(\rho+\mu)(\mu+d_1+\kappa))}{k_1\Delta(\alpha_1+\mu)(\rho+\mu) + \alpha_1 k_2\Delta(\rho+\mu) + k_4\Delta(\alpha_1+\mu)(\rho+\mu\varepsilon)}$.

$$
J_{\beta^*} = \begin{pmatrix}
-\mu & \alpha_2 & \rho & -\beta^* x_1^0 & \eta \\
0 & -(\alpha_1 + \mu) & 0 & 0 & 0 \\
0 & 0 & -(\rho + \mu) & -\beta^*\varepsilon x_3^0 & 0 \\
0 & 0 & 0 & \beta^* x_1^0 + \beta^*\varepsilon x_3^0 - (\mu + d_1 + \kappa) & 0 \\
0 & 0 & 0 & \kappa & -(\mu + \eta)
\end{pmatrix}.
$$

After some steps of the calculation we have determined the eigenvalues of $J_{\beta^*}$ as $\lambda_1 = -\mu$, $\lambda_2 = -(\alpha_1 + \mu)$ or or $\lambda_3 = -(\rho + \mu)$ or $\lambda_4 = 0$ or $\lambda_5 = -(\mu + \eta)$. It follows that the Jacobian $J(E^0_{CM})$ of Eq (14) at the disease-free equilibrium with $\beta_2 = \beta^*$, denoted by $J_{\beta^*}$, has a simple zero eigenvalue with all the remaining eigenvalues have negative real part. Hence, Theorem 2 of Castillo-Chavez and Song stated in [60] can be used to analyze the dynamics of the model to show that the model (11) undergoes backward bifurcation at $\mathcal{R}_C = 1$.

Eigenvectors of $J_{\beta^*}$: For the case $\mathcal{R}_C = 1$, it can be shown that the Jacobian of the system (14) at $\beta_2 = \beta^*$ (denoted by $J_{\beta^*}$) has a right eigenvectors associated with the zero eigenvalue

given by $u = (u_1, u_2, u_3, u_4, u_5)^T$ as

$$
\begin{pmatrix}
-\mu & \alpha_2 & \rho & -\beta^* x_1^0 & \eta \\
0 & -(\alpha_1 + \mu) & 0 & 0 & 0 \\
0 & 0 & -(\rho + \mu) & -\beta^* \varepsilon x_3^0 & 0 \\
0 & 0 & 0 & \beta^* x_1^0 + \beta^* \varepsilon x_3^0 - (\mu + d_1 + \kappa) & 0 \\
0 & 0 & 0 & \kappa & -(\mu + \eta)
\end{pmatrix}
\begin{pmatrix}
u_1 \\ u_2 \\ u_3 \\ u_4 \\ u_5
\end{pmatrix}
=
\begin{pmatrix}
0 \\ 0 \\ 0 \\ 0 \\ 0
\end{pmatrix}. \quad (15)
$$

Then solving Eq (15) the right eigenvectors associated with the zero eigenvalue are given by

$$
u_1 = \frac{-\beta^* x_1^0 u_4 (\rho + \mu)(\mu + \eta) - \beta^* \varepsilon x_3^0 \rho (\mu + \eta) u_4 + \kappa \eta (\rho + \mu) u_4}{\mu(\rho + \mu)(\mu + \eta)},
$$

$$
u_2 = 0, \; u_3 = -\frac{\beta^* \varepsilon x_3^0}{(\rho + \mu)} u_4, \; u_4 = u_4 > 0, \; u_5 = \frac{\kappa}{\mu + \eta} u_4.
$$

Similarly, the left eigenvector associated with the zero eigenvalues at $\beta_2 = \beta^*$ given by $v = (v_1, v_2, v_3, v_4, v_5)^T$ as

$$
\begin{pmatrix}
v_1 \\ v_2 \\ v_3 \\ v_4 \\ v_5
\end{pmatrix}^T
*
\begin{pmatrix}
-\mu & \alpha_2 & \rho & -\beta^* x_1^0 & \eta \\
0 & -(\alpha_1 + \mu) & 0 & 0 & 0 \\
0 & 0 & -(\rho + \mu) & -\beta^* \varepsilon x_3^0 & 0 \\
0 & 0 & 0 & D & 0 \\
0 & 0 & 0 & \kappa & -(\mu + \eta)
\end{pmatrix}
=
\begin{pmatrix}
0 \\ 0 \\ 0 \\ 0 \\ 0 \\ 0
\end{pmatrix}, \quad (16)
$$

where $D = \beta^* x_1^0 + \beta^* \varepsilon x_3^0 - (\mu + d_1 + \kappa)$.

Then solving Eq (16) the left eigenvectors associated with the zero eigenvalue are given by $v_1 = v_2 = v_3 = v_4 = 0$ and $v_4 = v_4 > 0$. After long steps of calculations the bifurcation coefficients $a$ and $b$ are obtained as

$$
a = \sum_{i,j,k=1}^{5} v_4 u_i u_j \frac{\partial^2 f_4}{\partial x_i \partial x_j} = 2v_4 u_1 u_4 \frac{\partial^2 f_4}{\partial x_1 \partial x_4} + 2v_4 u_3 u_4 \frac{\partial^2 f_4}{\partial x_3 \partial x_4}
$$

$$
= 2v_4 u_4 \left[ u_1 \frac{\partial^2 f_4}{\partial x_1 \partial x_4} + u_3 \frac{\partial^2 f_4}{\partial x_3 \partial x_4} \right], = 2v_4 u_4 [\beta_2 u_1 + \beta_2 \varepsilon u_3]
$$

$$
= 2v_4 u_4^2 \left[ \frac{-\beta_2 \beta^* x_1^0 (\rho + \mu)(\mu + \eta) - \beta_2 \beta^* \varepsilon x_3^0 \rho (\mu + \eta) + \beta_2 \kappa \eta (\rho + \mu) - \beta_2 \varepsilon \beta^* \varepsilon x_3^0 \mu (\mu + \eta)}{\mu(\rho + \mu)(\mu + \eta)} \right],
$$

$$
= 2v_4 u_4 [D_2 - D_1],
$$

where $D_1 = \frac{-\beta_2 \beta^* x_1^0 (\rho + \mu)(\mu + \eta) - \beta_2 \beta^* \varepsilon x_3^0 \rho (\mu + \eta) - \beta_2 \varepsilon \beta^* \varepsilon x_3^0 \mu (\mu + \eta)}{\mu(\rho + \mu)(\mu + \eta)}$, and $D_2 = \frac{\beta_2 \kappa \eta (\rho + \mu)}{\mu(\rho + \mu)(\mu + \eta)}$. Thus, the bifurcation coefficient $a$ is positive whenever $D_2 > D_1$.

Moreover

$$
b = \sum_{i,k=1}^{5} v_k u_i \frac{\partial^2 f_k}{\partial x_i \partial \beta} (E_{CM}^0) = \sum_{i=1}^{5} v_4 u_i \frac{\partial^2 f_4}{\partial x_i \partial \beta} = v_4 u_4 \frac{\partial^2 f_4}{\partial x_4 \partial \beta}
$$

$$
= v_4 u_4 [x_1^0 u_1 + \varepsilon x_3^0 u_3] > 0.
$$

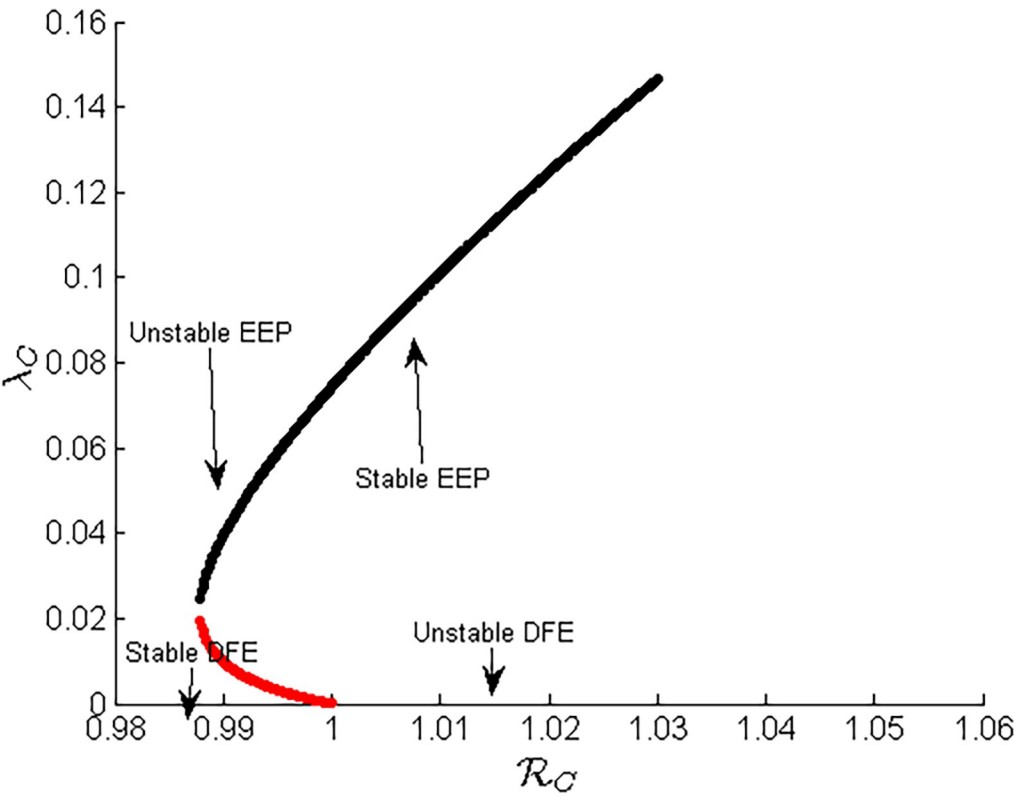

**Fig 2. Backward bifurcation diagram.**

Hence, from the theory of Castillo-Chavez and Song stated in [60] the COVID-19 mono-infection model (11) exhibits a phenomenon of backward bifurcation at $\mathcal{R}_C = 1$ and whenever $D_2 > D_1$.

The diagram representation of this bifurcation is given in Fig 2 below.

Fig 2 shows the appearance of backward bifurcation, which results in the coexistence of several equilibrium points. In such a case, the common conditions of disease eradication such as making $R_C < 1$ will not work, and the initial number of infected persons also plays a crucial role.

## 3.5. Analytical result of HIV/AIDS and COVID-19 co-infection model

### 3.5.1. Disease-free equilibrium point.
The disease free equilibrium point of the dynamical system (3) when the state variable $C_i = H_u = H_a = M_u = M_a = 0$ is given by $E_0 =$

$$\left( S^0,\ C_q^0,\ H_p^0,\ C_v^0,\ C_i^0,\ H_u^0, H_p^0,\ M_u^0,\ M_a^0, R^0,\ H_t^0 \right) =$$

$$\left( \frac{k_1\Delta}{\mu} + \frac{\alpha_1 k_2\Delta}{\alpha_1+\mu} + \frac{\alpha_2 k_3\Delta}{\alpha_2+\mu} + \frac{\rho k_4\Delta}{\rho+\mu}, \frac{k_2\Delta}{\alpha_1+\mu}, \frac{k_3\Delta}{\alpha_2+\mu},\ \frac{k_4\Delta}{\rho+\mu}, 0,\ 0,\ 0,\ 0,\ 0,\ 0,\ 0 \right).$$

### 3.5.2. Effective reproduction number of the co-infection model.
The effective reproduction number of the dynamical system (3) by applying the next generation operator method is the largest (dominant) eigenvalue (spectral radius) of the matrix: $FV^{-1} = \left[ \frac{\partial \mathcal{F}_i(E_0)}{\partial X_j} \right]\left[ \frac{\partial v_i(E_0)}{\partial X_j} \right],$ where $\mathcal{F}_i$ is the rate of appearance of new infection in compartment $i$, $v_i$ is the transfer of infections from one compartment $i$ to another, and $E_0$ is the disease-free equilibrium point. After

some steps of calculations we have determined that

$$
F = \begin{bmatrix}
\aleph_1 & 0 & \omega_1\aleph_1 & \omega_2\aleph_1 & 0 & 0 & 0 \\
0 & \aleph_2 & \rho_2\aleph_2 & \rho_3\aleph_2 & \rho_1\aleph_2 & 0 & 0 \\
0 & 0 & 0 & 0 & 0 & 0 & 0 \\
0 & 0 & 0 & 0 & 0 & 0 & 0 \\
0 & 0 & 0 & 0 & 0 & 0 & 0 \\
0 & 0 & 0 & 0 & 0 & 0 & 0 \\
0 & 0 & 0 & 0 & 0 & 0 & 0
\end{bmatrix},
$$

where $\aleph_1 = \beta_2 S + \beta_2 H_p + \varepsilon\beta_2 C_v$, $\aleph_2 = \frac{\beta_1}{N}\left(S + C_q + C_v\right)$, and

$$
V = \begin{bmatrix}
(\mu + d_1 + \kappa) & 0 & 0 & 0 & 0 & 0 & 0 \\
0 & (\theta + \mu + d_2) & -\theta_1 & 0 & 0 & 0 & 0 \\
0 & 0 & (\mu + d_4 + \delta + \theta_1) & 0 & 0 & 0 & 0 \\
0 & 0 & -\delta & (\mu + d_5 + \theta_2) & 0 & 0 & 0 \\
0 & -\theta & 0 & -\theta_2 & (\gamma + d_3 + \mu) & 0 & 0 \\
-\kappa & 0 & 0 & 0 & 0 & (\mu + \eta) & 0 \\
0 & 0 & 0 & 0 & -\gamma & 0 & \mu
\end{bmatrix}.
$$

Applying Mathematica we have determined as

$$
FV^{-1} = \begin{bmatrix}
\dfrac{\aleph_1}{(\mu + d_1 + \kappa)} & a_{21} & 0 & 0 & 0 & 0 & 0 \\
0 & \dfrac{\aleph_2}{(\theta + \mu + d_2)} + \dfrac{\rho_1\aleph_2\theta}{(\theta + \mu + d_2)(\gamma + d_3 + \mu)} & 0 & 0 & 0 & 0 & 0 \\
0 & a_{23} & 0 & 0 & 0 & 0 & 0 \\
0 & a_{24} & 0 & 0 & 0 & 0 & 0 \\
0 & a_{25} & 0 & 0 & 0 & 0 & 0 \\
0 & a_{26} & 0 & 0 & 0 & 0 & 0 \\
0 & a_{27} & 0 & 0 & 0 & 0 & 0
\end{bmatrix}.
$$

After some computations and simplifications we have determined the dominant eigenvalue in magnitude of the matrix $FV^{-1}$ which is the HIV/AIDS and COVID-19 co-infection effective reproduction number given by

$$
\mathcal{R}_0^{HC} = \max\{\mathcal{R}_C,\ \mathcal{R}_{HM}\} = \max\{\frac{\beta_2 k_1\Delta(\alpha_1 + \mu)(\rho + \mu) + \beta_2\alpha_1 k_2\Delta(\rho + \mu) + \beta_2 k_4\Delta\rho(\alpha_1 + \mu) + \beta_2\varepsilon k_4\Delta\mu(\alpha_1 + \mu)}{\mu(\alpha_1 + \mu)(\rho + \mu)(\mu + d_1 + \kappa)}
$$

$$
,\frac{\beta_1(1 - k_3)(\alpha_2 + \mu) + \beta_1\alpha_2 k_3}{(\alpha_2 + \mu)(\theta + \mu + d_2)} + \frac{\beta_1\rho_1\theta(1 - k_3)(\alpha_2 + \mu) + \beta_1\rho_1\theta\alpha_2 k_3}{(\theta + \mu + d_2)(\gamma + \mu + d_3)}\},\text{where } \mathcal{R}_C =
$$

$\frac{\beta_2 S^0 + \varepsilon\beta_2 C_v^0}{\mu + d_1 + \kappa} = \frac{\beta_2 k_1\Delta(\alpha_1 + \mu)(\rho + \mu) + \beta_2\alpha_1 k_2\Delta(\rho + \mu) + \beta_2 k_4\Delta\rho(\alpha_1 + \mu) + \beta_2\varepsilon k_4\Delta\mu(\alpha_1 + \mu)}{\mu(\alpha_1 + \mu)(\rho + \mu)(\mu + d_1 + \kappa)}$ is the COVID-19 effective reproduction number and $\mathcal{R}_{HM} = \frac{\beta_1(1 - k_3)(\alpha_2 + \mu) + \beta_1\alpha_2 k_3}{(\alpha_2 + \mu)(\theta + \mu + d_2)} + \frac{\beta_1\rho_1\theta(1 - k_3)(\alpha_2 + \mu) + \beta_1\rho_1\theta\alpha_2 k_3}{(\theta + \mu + d_2)(\gamma + \mu + d_3)}$ is the HIV/AIDS effective reproduction number.

**3.5.3. Locally asymptotically stability of the disease-free equilibrium point.** The Jacobian matrix of the system (3) at disease free equilibrium point is given as

$$
J(E_0) = \begin{pmatrix}
-\mu & \alpha_1 & \alpha_2 & \rho & -\beta_2 A_1 & -\frac{\beta_1}{N}A_1 & -\frac{\beta_1}{N}A_1\rho_1 & -\aleph_4 & -\aleph_6 & \eta & 0 \\
0 & -(\alpha_1+\mu) & 0 & 0 & 0 & -\frac{\beta_1}{N}A_2 & -\frac{\beta_1}{N}A_2\rho_1 & -\frac{\beta_1}{N}A_2\rho_2 & -\frac{\beta_1}{N}A_2\rho_3 & 0 & 0 \\
0 & 0 & -(\alpha_2+\mu) & 0 & -\beta_2 A_3 & 0 & 0 & -\beta_2 A_3\omega_1 & -\beta_2 A_3\omega_2 & 0 & 0 \\
0 & 0 & 0 & -(\rho+\mu) & -\varepsilon\beta_2 A_4 & -\frac{\beta_1}{N}A_4 & -\frac{\beta_1}{N}A_4\rho_1 & -\aleph_5 & -\aleph_7 & 0 & 0 \\
0 & 0 & 0 & 0 & \aleph_3 & 0 & 0 & \omega_1\aleph_3 & \omega_2\aleph_3 & 0 & 0 \\
0 & 0 & 0 & 0 & 0 & \aleph_8 & \rho_1\aleph_8 & \rho_2\aleph_8+\theta_1 & \rho_3\aleph_8 & 0 & 0 \\
0 & 0 & 0 & 0 & 0 & \Theta & \aleph_9 & 0 & \theta_2 & 0 & 0 \\
0 & 0 & 0 & 0 & 0 & 0 & 0 & \aleph_{10} & 0 & 0 & 0 \\
0 & 0 & 0 & 0 & 0 & 0 & 0 & \delta & \aleph_{11} & 0 & 0 \\
0 & 0 & 0 & 0 & \kappa & 0 & 0 & 0 & 0 & -(\mu+\eta) & 0 \\
0 & 0 & 0 & 0 & 0 & 0 & \gamma & 0 & 0 & 0 & -\mu
\end{pmatrix},
$$

where $\aleph_3 = \beta_2\left(S^0 + H_p + \varepsilon C_v\right) - (\mu+d_1+\kappa)$, $\aleph_4 = \left(\frac{\beta_1}{N}\rho_2 + \beta_2\omega_1\right)S^0$, $\aleph_5 = \left(\frac{\beta_1}{N}\rho_2 + \varepsilon\beta_2\omega_1\right)C_v$, $\aleph_6 = \left(\frac{\beta_1}{N}\rho_3 + \beta_2\omega_2\right)S^0$ and $\aleph_7 = \left(\frac{\beta_1}{N}\rho_3 + \varepsilon\beta_2\omega_2\right)C_v$, $\aleph_8 = \frac{\beta_1}{N}\left(S^0 + C_q + C_v\right) - (\theta+\mu+d_2)$, $\aleph_9 = -(\gamma+d_3+\mu)$, $\aleph_{10} = -(\mu+d_4+\delta+\theta_1)$, $\aleph_{11} = -(\mu+d_5+\theta_2)$.

Then the eigenvalues of the matrix $J(E_0)$ are $\lambda_1 = -\mu < 0$ or $\lambda_2 = -(\alpha_1+\mu) < 0$ or $\lambda_3 = -(\alpha_2+\mu) < 0$ or $\lambda_4 = -(\rho+\mu) < 0$ or $\lambda_5 = -\mu < 0$ or $\lambda_6 = -(\mu+\eta) < 0$ or $\lambda_7 = \frac{\beta_2\varepsilon k_4\Delta}{(\rho+\mu)(\mu+d_1+\kappa)}\left(\mathcal{R}_{CM} - 1\right) < 0$ or $\lambda_8 = -(\mu+d_4+\delta+\theta_1) < 0$ or $\lambda_9 = -(\mu+d_5+\Theta_2) < 0$ or $\lambda^2 + [(\gamma+d_3+\mu) + (\theta+\mu+d_2) - \aleph_8]\lambda - [(\aleph_8 - (\theta+\mu+d_2))(\gamma+d_3+\mu) + \theta\rho_1\aleph_8] = 0$.

Then after some calculations we have got the last two eigenvalues of the quadratic equation as $\lambda_{10} < 0$ and $\lambda_{11} < 0$ whenever $\mathcal{R}_0^{HC} = \max\{\mathcal{R}_C, \mathcal{R}_{HM}\} < 1$. Thus, since all the eigenvalues are negative, the disease-free equilibrium point of the full model (3) is locally asymptotically stable whenever $\mathcal{R}_0^{HC} = \max\{\mathcal{R}_C, \mathcal{R}_{HM}\} < 1$.

**3.5.4. Global asymptotic stability of disease-free equilibrium point.** In this sub-section we have used the method derived by Castillo-Chavez et al. and stated in reference [61] to look into the global asymptotic stability (GAS) of the co-infection model (3) disease-free equilibrium point. We mention two requirements that, if satisfied, also ensure the disease-free equilibrium is globally asymptotically stable. Then the new system (3) is rewritten as:

$$
\frac{d\Psi}{dt} = F(\Psi, Y),
$$
$$
\frac{dY}{dt} = G(\Psi, Y), \ G(\Psi, 0) = 0,
$$

where $\Psi = (S, C_q, H_p, C_v) \in \mathbb{R}^4$ denotes the number of uninfected components and $z =$

$(C_i, H_u, H_a, M_u, M_a, R, H_t) \in \mathbb{R}^7$ denotes the number of infected components. $\Pi_0 = (\Psi_0, 0)$, denotes the disease-free equilibrium point of the system. The following requirements must be satisfied to ensure the globally asymptotic stability:

$(H_1)$ For $\frac{d\Psi}{dt} = F(\Psi, 0)$, $\Pi_0$ is globally asymptotically stable.

$(H_2)$ $G(\Psi, Y) = AY - \widehat{G}(\Psi, Y)$, $\widehat{G}(\Psi, Y) \geq 0$, for $(\Psi, Y) \in \Omega$, where $A = D_Y G(\Psi_0, 0)$ is a Metzler matrix (the off diagonal elements of $A$ are nonnegative) and $\Omega$ is the region where the model makes biological sense.

**Theorem 8**: The fixed point $\Pi_0 = (\Psi_0, 0)$ is a globally asymptotically stable equilibrium point of system (3) provided $\mathcal{R}_0^{HC} < 1$ and the assumptions $(H_1)$ and $(H_2)$ are satisfied otherwise unstable.

**Proof**: The system (1) is rewritten as

$$\frac{d\Psi}{dt} = F(\Psi, Y) = \begin{pmatrix} k_1\Delta + \alpha_1 C_q + \alpha_2 H_p + \rho C_v + \eta R - (\lambda_H + \lambda_C + \mu)S \\ k_2\Delta - (\lambda_H + \alpha_1 + \mu)C_q \\ k_3\Delta - (\alpha_2 + \mu + \lambda_C)H_p \\ k_4\Delta - (\rho + \mu + \lambda_H + \varepsilon\lambda_C)C_v \end{pmatrix},$$

$$F(\Psi, 0) = \begin{pmatrix} k_1\Delta + \alpha_1 C_q + \alpha_2 H_p + \rho C_v - \mu S \\ k_2\Delta - (\alpha_1 + \mu)C_q \\ k_3\Delta - (\alpha_2 + \mu)H_p \\ k_4\Delta - (\rho + \mu)C_v \end{pmatrix},$$

where $\Psi$ represents the number of non-infectious compartments and $Y$ represents the number of infectious compartments.

And

$$G(\Psi, Y) = \begin{pmatrix} \lambda_C S + \lambda_C H_P + \varepsilon\lambda_C C_v - (\mu + d_1 + \kappa + \upsilon\lambda_H)C_i \\ \lambda_H S + \lambda_H C_q + \lambda_H C_v + \theta_1 M_u - (\theta + \mu + d_2 + \phi_1\lambda_C)H_u \\ \theta H_u + \theta_2 M_a + \theta_3 M_u - (\gamma + d_3 + \mu + \phi_2\lambda_C)H_a \\ \phi_1\lambda_C H_u + \upsilon\lambda_H C_i - (\mu + \theta_3 + d_4 + \delta + \Theta_1)M_u \\ \delta M_u + \phi_2\lambda_C H_a - (\mu + d_5 + \theta_2)M_a \\ \kappa C_i - (\mu + \eta)R \\ \gamma H_a - \mu H_t \end{pmatrix}.$$

Then $G(\Psi, Y) = AY - \widehat{G}(\Psi, Y)$, where,

$$
A = \begin{pmatrix}
\Sigma_1 & 0 & 0 & \beta_2\omega_1(S + H_P + \varepsilon C_v) & \beta_2\omega_2(S + H_P + \varepsilon C_v) & 0 & 0 \\
0 & \Sigma_2 & \Sigma_3 & \frac{\beta_1\rho_2}{N}\left(S + C_q + C_v\right) + \theta_1 & \frac{\beta_1\rho_3}{N}\left(S + C_q + C_v\right) & 0 & 0 \\
0 & \theta & \Sigma_4 & \theta_3 & \theta_2 & 0 & 0 \\
0 & 0 & 0 & -(\mu + \theta_3 + d_4 + \delta + \Theta_1) & 0 & 0 & 0 \\
0 & 0 & 0 & \delta & -(\mu + d_5 + \theta_2) & 0 & 0 \\
\kappa & 0 & 0 & 0 & 0 & -(\mu + \eta) & 0 \\
0 & 0 & \gamma & 0 & 0 & 0 & -\mu
\end{pmatrix},
$$

where $\Sigma_1 = \beta_2(S + H_P + \varepsilon C_v) - (\mu + d_1 + \kappa)$, $\Sigma_2 = \frac{\beta_1}{N}\left(S + C_q + C_v\right) - (\theta + \mu + d_2)$, $\Sigma_3 = \frac{\beta_1\rho_1}{N}\left(S + C_q + C_v\right)$ and $\Sigma_4 = -(\gamma + d_3 + \mu)$, so that

$$
AY = \begin{pmatrix}
\lambda_C(S + H_P + \varepsilon C_v) - (\mu + d_1 + \kappa)C_i \\
-(\theta + \mu + d_2)H_u + \lambda_H\left(S + C_q + C_v\right) + \theta_1 M_u \\
\theta H_u - (\gamma + d_3 + \mu)H_a + \theta_3 M_u + \theta_2 M_a \\
-(\mu + \theta_3 + d_4 + \delta + \Theta_1)M_u \\
\delta M_u - (\mu + d_5 + \theta_2)M_a \\
\kappa C_i - (\mu + \eta)R \\
\gamma H_a - \mu H_t
\end{pmatrix}.
$$

We have determined that,

$$
\widehat{G}(\Psi, Y) = \begin{pmatrix}
\lambda_C\frac{\Delta}{\mu}\left(k_1 + \frac{\alpha_1 k_2\mu}{\alpha_1 + \mu} + \frac{(\alpha_2 + 1)k_3\mu}{\alpha_2 + \mu} + \frac{(\rho + 1)k_4\mu}{\rho + \mu}\right) - \lambda_C\left(S + C_q + \varepsilon C_v\right) + \upsilon\lambda_H C_i \\
\lambda_H\frac{\Delta}{\mu}\left(k_1 + \frac{k_2\mu(\alpha_1 + 1)}{\alpha_1 + \mu} + \frac{\alpha_2 k_3\mu}{\alpha_2 + \mu} + \frac{k_4\mu(\rho + 1)}{\rho + \mu}\right) - \lambda_H\left(S + C_q + C_v\right) + \phi_1\lambda_C H_u \\
\phi_2\lambda_C H_a \\
-\phi_1\lambda_C H_u - \upsilon\lambda_H C_i \\
-\phi_2\lambda_C H_a \\
0 \\
0
\end{pmatrix}.
$$

It is clear from the above discussion, that, $\widehat{G}(\Psi, Y) \not\geq 0$. Hence by the same reason given by results in reference [38], the disease-free equilibrium point may not be globally asymptotically stable.

## 4. Analysis of the optimal control strategy

In this section, we provide a thorough qualitative analysis of the time-dependent HIV/AIDS and COVID-19 co-infection model (3). The Pontryagin's Maximum Principle stated in literatures [25,43,51,52,55] is used to describe this analysis, with the aim of minimizing the HIV/AIDS infection aware individuals denoted by $H_a$, the COVID-19 infected individuals denoted by $C_i$ and the total HIV/AIDS and COVID-19 co-infected individuals denoted by $M_u + M_a$. In the case of time-dependent optimal control, we employ Pontryagin's Maximum Principle to derive the necessary conditions for diseases control mechanisms. After incorporating the controls into the HIV/AIDS and COVID-19 co-infection transmission model (3), the optimal control problem is as follows:

$$\dot{S} = k_1\Delta + \alpha_1 C_q + \alpha_2 H_p + \rho C_v + \eta R - (1 - \mathfrak{u}_1)\lambda_H S - (1 - \mathfrak{u}_2)\lambda_C S - \mu S,$$

$$\dot{C}_q = k_2\Delta - (1 - \mathfrak{u}_1)\lambda_H C_q - (\alpha_1 + \mu)C_q,$$

$$\dot{H}_p = k_3\Delta - (1 - \mathfrak{u}_2)\lambda_C H_p - (\alpha_2 + \mu)H_p,$$

$$\dot{C}_v = k_4\Delta - (1 - \mathfrak{u}_1)\lambda_H C_v - (1 - \mathfrak{u}_2)\varepsilon\lambda_C C_v - (\rho + \mu)C_v,$$

$$\dot{C}_i = (1 - \mathfrak{u}_2)\lambda_C S + (1 - \mathfrak{u}_2)\lambda_C H_p + (1 - \mathfrak{u}_2)\varepsilon\lambda_C C_v - (1 - \mathfrak{u}_1)\upsilon\lambda_H C_i - (\mu + d_1 + \mathfrak{u}_3\kappa)C_i,$$

$$\dot{H}_u = (1 - \mathfrak{u}_1)\lambda_H S + (1 - \mathfrak{u}_1)\lambda_H C_q + (1 - \mathfrak{u}_1)\lambda_H C_v + \mathfrak{u}_3\theta_1 M_u - (1 - \mathfrak{u}_2)\phi_1\lambda_C H_u - (\theta + \mu + d_2)H_u, \quad (17)$$

$$\dot{H}_a = \theta H_u + \mathfrak{u}_3\theta_2 M_a - (1 - \mathfrak{u}_2)\phi_2\lambda_C H_a - (\mathfrak{u}_4\gamma + d_3 + \mu)H_a,$$

$$\dot{M}_u = (1 - \mathfrak{u}_2)\phi_1\lambda_C H_u + (1 - \mathfrak{u}_1)\upsilon\lambda_H H_p - (\mu + d_4 + \delta + \mathfrak{u}_3\theta_1)M_u,$$

$$\dot{M}_a = \delta M_u + (1 - \mathfrak{u}_2)\phi_2\lambda_C H_p - (\mu + d_5 + \mathfrak{u}_3\theta_2)M_a,$$

$$\dot{R} = \mathfrak{u}_3\kappa A_5 - (\mu + \eta)R,$$

$$\dot{H}_t = \mathfrak{u}_4\gamma H_p - \mu H_t,$$

with the corresponding initial conditions

$$S(0) > 0, C_q(0) \geq 0, H_p(0) \geq 0, C_v(0) \geq 0, C_i(0) \geq 0, H_u(0) \geq 0, H_a(0) \geq 0, M_u(0)$$
$$\geq 0, M_a(0) \geq 0, R(0) > 0, \text{ and } H_t(0) > 0, \quad (18)$$

and $0 \leq \mathfrak{u}_1(t) \leq 1$ represents HIV/AIDS infection protective control, $0 \leq \mathfrak{u}_2(t) \leq 1$ represents the COVID-19 infections protective control using quarantine, $0 \leq \mathfrak{u}_3(t) \leq 1$ represents the COVID-19 infection treatment control, and $0 \leq \mathfrak{u}_4(t) \leq 1$ represents the HIV/AIDS treatment control.

The objective is to find the optimal control values $\mathfrak{u}^* = \left(\mathfrak{u}_1^*, \mathfrak{u}_2^*, \mathfrak{u}_3^*, \mathfrak{u}_4^*\right)$ of the controls $\mathfrak{u} = (\mathfrak{u}_1, \mathfrak{u}_2, \mathfrak{u}_3, \mathfrak{u}_4)$ such that the associated state trajectories $\left(S^*, C_q^*, H_p^*, C_v^*, C_i^*, H_u^*, H_p^*, M_u^*, M_a^*, R^*, H_t^*\right)$ are solution of the optimal control system (17) in the intervention time interval $[0, T_f]$ with initial conditions as given in (18) and minimize

the objective functional given by

$$J(\mathfrak{u}_1,\ \mathfrak{u}_2, \mathfrak{u}_3,\ \mathfrak{u}_4)$$
$$= \int_0^{T_f} \left( \mathfrak{w}_1 C_i + \mathfrak{w}_2\, H_p + \mathfrak{w}_3 M_u + \mathfrak{w}_4\, M_a + \frac{\mathfrak{B}_1}{2}\mathfrak{u}_1^2 + \frac{\mathfrak{B}_2}{2}\mathfrak{u}_2^2 + \frac{\mathfrak{B}_3}{2}\mathfrak{u}_3^2 + \frac{\mathfrak{B}_4}{2}\mathfrak{u}_4^2 \right) dt, \quad (19)$$

where the coefficients $\mathfrak{w}_1, \mathfrak{w}_2,\ \mathfrak{w}_3,$
and $\mathfrak{w}_4$ are positive weight constants and $\frac{\mathfrak{B}_1}{2},\ \frac{\mathfrak{B}_2}{2},\ \frac{\mathfrak{B}_3}{2}\ $ and $\frac{\mathfrak{B}_4}{2}$ are the measure of relative costs of interventions associated with the controls $\mathfrak{u}_1,\ \mathfrak{u}_2, \mathfrak{u}_3$ and $\ \mathfrak{u}_4$, respectively, and also balances the units of integrand. In the cost functional, the term $\mathfrak{w}_1 A_5$ refer to the cost related to COVID-19 infected class, the term $\mathfrak{w}_2\, H_u$ refer to the cost related to individuals mono-infected with HIV and aware, the term $\mathfrak{w}_3 A_8$ refer to the cost related to co-infected individuals unaware of HIV infection and the term $\mathfrak{w}_4 M_a$ refer to the cost related to co-infected individuals aware of HIV infection.

$$I\Big(S,\ C_q, H_p, C_v,\ C_i,\ H_u,\ H_a,\ M_u, M_a, R,\ H_t, \mathfrak{u}\Big) = \mathfrak{w}_1,\ C_i + \mathfrak{w}_2\, H_a + \mathfrak{w}_3 M_u + \mathfrak{w}_4\, M_a +$$
$\frac{\mathfrak{B}_1}{2}\mathfrak{u}_1^2 + \frac{\mathfrak{B}_2}{2}\mathfrak{u}_2^2 + \frac{\mathfrak{B}_3}{2}\mathfrak{u}_3^2 + \frac{\mathfrak{B}_4}{2}\mathfrak{u}_4^2$, measures the current cost at time t. The set of admissible Lebesgue measurable control functions is defined by

$$\Omega_{\mathfrak{u}} = \left\{ (\mathfrak{u}_1(t),\ \mathfrak{u}_2(t), \mathfrak{u}_3(t),\ \mathfrak{u}_4(t)) \in L^4 : 0 \le \mathfrak{u}_1(t),\ \mathfrak{u}_2(t), \mathfrak{u}_3(t),\ \mathfrak{u}_4(t) \le 1,\ t \in \left[0, T_f\right] \right\}. (20)$$

More precisely, we seek an optimal control pair

$$J\Big(\mathfrak{u}_1^*,\ \mathfrak{u}_2^*, \mathfrak{u}_3^*,\ \mathfrak{u}_4^*\Big) = \min_{\Omega_{\mathfrak{u}}} J(\mathfrak{u}_1,\ \mathfrak{u}_2, \mathfrak{u}_3,\ \mathfrak{u}_4). \quad (21)$$

**Theorem 9 (Existence Theorem)**: There exists an optimal control $\mathfrak{u}^* = \Big(\mathfrak{u}_1^*,\ \mathfrak{u}_2^*, \mathfrak{u}_3^*,\ \mathfrak{u}_4^*\Big)$ in $\Omega_{\mathfrak{u}}$ and a corresponding solution vector $\Big(S^*,\ C_q^*,\ H_p^*, C_v^*,\ C_i^*,\ H_u^*,\ H_a^*,\ M_u^*,\ M_a^*,\ R^*,\ H_t^*\Big)$ to the optimal control dynamical system (17) with the initial values (18) such that $J\Big(\mathfrak{u}_1^*,\ \mathfrak{u}_2^*, \mathfrak{u}_3^*,\ \mathfrak{u}_4^*\Big) = \min_{\Omega_{\mathfrak{u}}} J(\mathfrak{u}_1,\ \mathfrak{u}_2, \mathfrak{u}_3,\ \mathfrak{u}_4)$.

<u>Note</u>: We utilize Pontryagin's Maximal principle stated in literatures [51,52,55], to determine the prerequisites for the optimal control model (17). The optimal control problem (17) and (19) defined Hamiltonian (H) function is expressed as

$$\mathcal{H} = \mathfrak{w}_1 C_i + \mathfrak{w}_2\, H_a + \mathfrak{w}_3 M_u + \mathfrak{w}_4\, M_a + \frac{\mathfrak{B}_1}{2}\mathfrak{u}_1^2 + \frac{\mathfrak{B}_2}{2}\mathfrak{u}_2^2 + \frac{\mathfrak{B}_3}{2}\mathfrak{u}_3^2 + \frac{\mathfrak{B}_4}{2}\mathfrak{u}_4^2 + \sum_{i=1}^{11} \lambda_i \mathcal{G}_i, \quad (22)$$

where $\mathcal{G}_i$ stands for the $i^{th}$ state variable equation and $\lambda_1(t), \lambda_2(t), \lambda_3(t), \lambda_4(t), \lambda_5(t), \lambda_6(t), \lambda_7(t),$ $\lambda_8(t), \lambda_9(t), \lambda_{10}(t)$ and $\lambda_{11}(t)$ are adjoint variables. Similarly to obtain the co-state variables by using Pontryagin's Maximum Principle stated in literatures [51,52,55], with the existence result the following theorem is stated:

**Theorem 10**: Let $\mathfrak{u}^* = \Big(\mathfrak{u}_1^*,\ \mathfrak{u}_2^*, \mathfrak{u}_3^*,\ \mathfrak{u}_4^*\Big)$ be the optimal control and $S^*,\ C_q^*,\ H_p^*, C_v^*,\ C_i^*,\ H_u^*,\ H_a^*,\ M_u^*,\ M_a^*,\ R^*,\ H_t^*$ be the associated unique optimal solutions of the optimal control problem (17) with initial condition (18) and objective functional (19) with fixed final time $T_f$ (20). Then there exists adjoint function $\lambda_i^*(\cdot),\ i = 1, \ldots, 11$ satisfying the

following canonical equations

$$\frac{d\lambda_1}{dt} = (1-\mathfrak{u}_1)\lambda_H^*(\lambda_1 - \lambda_6) + (1-\mathfrak{u}_2)\lambda_C^*(\lambda_1 - \lambda_5) + \mu\lambda_1,$$

$$\frac{d\lambda_2}{dt} = (1-\mathfrak{u}_1)\lambda_H^*(\lambda_2 - \lambda_6) + \alpha_1(\lambda_2 - \lambda_1) + \mu\lambda_2,$$

$$\frac{d\lambda_3}{dt} = (1-\mathfrak{u}_2)\lambda_C^*(\lambda_3 - \lambda_5) + \alpha_2(\lambda_3 - \lambda_1) + \mu\lambda_3,$$

$$\frac{d\lambda_4}{dt} = (1-\mathfrak{u}_1)\lambda_H^*(\lambda_4 - \lambda_6) + (1-\mathfrak{u}_2)\varepsilon\lambda_C^*(\lambda_4 - \lambda_5) + \rho(\lambda_4 - \lambda_1) + \mu\lambda_4,$$

$$\frac{d\lambda_5}{dt} = -\mathfrak{w}_1 + (1-\mathfrak{u}_2)\beta_2 A_1^*(\lambda_1 - \lambda_5) + (1-\mathfrak{u}_2)\beta_2 A_3^*(\lambda_3 - \lambda_5) + (1-\mathfrak{u}_2)\varepsilon\beta_2 A_4^*(\lambda_4 - \lambda_5) +$$
$$(1-\mathfrak{u}_2)\phi_1\beta_2 A_6^*(\lambda_6 - \lambda_8) + (1-\mathfrak{u}_2)\phi_2\beta_2 A_7^*(\lambda_7 - \lambda_9) + (1-\mathfrak{u}_1)\upsilon\lambda_H(\lambda_5 - \lambda_8) + (\mu + d_1)\lambda_5 + \mathfrak{u}_3\kappa(\lambda_5 - \lambda_{10}),$$

$$\frac{d\lambda_6}{dt} = (1-\mathfrak{u}_1)\frac{\beta_1}{N}A_1^*(\lambda_1 - \lambda_6) + (1-\mathfrak{u}_1)\frac{\beta_1}{N}A_2^*(\lambda_2 - \lambda_6) + (1-\mathfrak{u}_1)\frac{\beta_1}{N}A_4^*(\lambda_4 - \lambda_6) +$$
$$(1-\mathfrak{u}_1)\upsilon\frac{\beta_1}{N}A_5^*(\lambda_5 - \lambda_8) + (1-\mathfrak{u}_2)\phi_1\lambda_C^*(\lambda_6 - \lambda_8) + (\mu + d_2)\lambda_6 + \Theta(\lambda_6 - \lambda_7),$$

$$\frac{d\lambda_7}{dt} = -\mathfrak{w}_2 + (1-\mathfrak{u}_1)\frac{\beta_1\rho_1}{N}A_1^*(\lambda_1 - \lambda_6) + (1-\mathfrak{u}_1)\frac{\beta_1\rho_1}{N}A_2^*(\lambda_2 - \lambda_6) + (1-\mathfrak{u}_1)\frac{\beta_1\rho_1}{N}A_4^*(\lambda_4 - \lambda_6) +$$
$$(1-\mathfrak{u}_1)\upsilon\frac{\beta_1\rho_1}{N}A_5^*(\lambda_5 - \lambda_8) + (1-\mathfrak{u}_2)\phi_2\lambda_C^*(\lambda_7 - \lambda_9) + (d_3 + \mu)\lambda_7 + \mathfrak{u}_4\gamma(\lambda_7 - \lambda_{11}),$$

$$\frac{d\lambda_8}{dt} = -\mathfrak{w}_3 + (1-\mathfrak{u}_1)\frac{\beta_1\rho_2}{N}A_1^*(\lambda_1 - \lambda_6) + (1-\mathfrak{u}_2)\beta_2\omega_1 A_1^*(\lambda_1 - \lambda_5) + (1-\mathfrak{u}_1)\frac{\beta_1\rho_2}{N}A_2^*(\lambda_2 - \lambda_6) + (1-\mathfrak{u}_2)$$
$$\beta_2\omega_1 A_3^*(\lambda_3 - \lambda_5) + (1-\mathfrak{u}_1)\frac{\beta_1\rho_2}{N}A_4^*(\lambda_4 - \lambda_6) + (1-\mathfrak{u}_2)\varepsilon\beta_2\omega_1 A_4^*(\lambda_4 - \lambda_5) + (1-\mathfrak{u}_1)\upsilon\frac{\beta_1\rho_2}{N}A_5^*(\lambda_5 - \lambda_8) + \mathfrak{u}_3\theta_1$$
$$(\lambda_8 - \lambda_6) + (1-\mathfrak{u}_2)\phi_1\beta_2\omega_1 A_6^*(\lambda_6 - \lambda_8) + (1-\mathfrak{u}_2)\phi_2\beta_2\omega_1 A_7^*(\lambda_7 - \lambda_9) + (\mu + d_4)\lambda_8 + \delta(\lambda_8 - \lambda_9),$$

$$\frac{d\lambda_9}{dt} = -\mathfrak{w}_4 + (1-\mathfrak{u}_1)\frac{\beta_1\rho_3}{N}A_1^*(\lambda_1 - \lambda_6) + (1-\mathfrak{u}_2)\beta_2\omega_2 A_1^*(\lambda_1 - \lambda_5) + (1-\mathfrak{u}_1)\frac{\beta_1\rho_3}{N}A_2^*(\lambda_2 - \lambda_6) + (1-\mathfrak{u}_2)\beta_2\omega_2 A_3^*$$
$$(\lambda_3 - \lambda_5) + (1-\mathfrak{u}_1)\frac{\beta_1\rho_3}{N}A_4^*(\lambda_4 - \lambda_6) + (1-\mathfrak{u}_2)\varepsilon\beta_2\omega_2 A_4^*(\lambda_4 - \lambda_5) + (1-\mathfrak{u}_1)\upsilon\frac{\beta_1\rho_3}{N}A_5^*(\lambda_5 - \lambda_8) +$$
$$(1-\mathfrak{u}_2)\phi_1\beta_2\omega_2 A_6^*(\lambda_6 - \lambda_8) + \mathfrak{u}_3\theta_2(\lambda_9 - \lambda_7) + (1-\mathfrak{u}_2)\phi_2\beta_2\omega_2 A_7^*(\lambda_7 - \lambda_9) + (\mu + d_5)\lambda_9,$$

$$\frac{d\lambda_{10}}{dt} = -\eta\lambda_1 + (\mu + \eta)\lambda_{10},$$

$$\frac{d\lambda_{11}}{dt} = \mu\lambda_{11},$$

$$(23)$$

with transiversality conditions

$$\lambda_i^*\left(T_f\right) = 0, \ i = 1, \ 2, \ \ldots, 11. \tag{24}$$

Moreover, the corresponding optimal controls $\mathfrak{u}_1^*(t)$, $\mathfrak{u}_2^*(t)$, $\mathfrak{u}_3^*(t)$, and $\mathfrak{u}_4^*(t)$ are given by

$$\mathfrak{u}_1^*(t) = \max\left\{0, \min\left\{\frac{\lambda_H^* S^*(\lambda_6 - \lambda_1) + \lambda_H^* C_q^*(\lambda_6 - \lambda_2) + \lambda_H^* C_v^*(\lambda_6 - \lambda_4) + \upsilon\lambda_H^* C_i^*(\lambda_8 - \lambda_5)}{\mathfrak{B}_1}, 1\right\}\right\},$$

$$\mathfrak{u}_2^*(t) = \max\left\{0, \min\left\{\frac{\lambda_C^* S^*(\lambda_5 - \lambda_1) + \lambda_C^* H_p^*(\lambda_5 - \lambda_3) + \varepsilon\lambda_C^* C_v^*(\lambda_5 - \lambda_4) + \phi_1\lambda_C^* H_u^*(\lambda_8 - \lambda_6) + \phi_2\lambda_C^* H_p^*(\lambda_9 - \lambda_7)}{\mathfrak{B}_2}, 1\right\}\right\},$$

$$\mathfrak{u}_3^*(t) = \max\left\{0, \min\left\{\frac{\Theta_1 M_u^*(\lambda_8 - \lambda_6) + \Theta_2 M_a^*(\lambda_9 - \lambda_7) + \kappa C_i^*(\lambda_5 - \lambda_{10})}{\mathfrak{B}_3}, 1\right\}\right\},$$

$$\mathfrak{u}_4^*(t) = \max\left\{0, \min\left\{\frac{\gamma H_p^*(\lambda_7 - \lambda_{11})}{\mathfrak{B}_4}, 1\right\}\right\}.$$

$$(25)$$

**Proof**: To obtain the form of the co-state equations we compute the derivative of the Hamiltonian function ($\mathcal{H}$), given in (22), with respect to $S^*$, $C_q^*$, $H_p^*$, $C_v^*$, $C_i^*$, $H_u^*$, $H_a^*$, $M_u^*$, $M_a^*$, $R^*$ and $H_t^*$ respectively. Then the adjoint or co-state equations obtained are given by:

$$\frac{d\lambda_1}{dt} = -\frac{\partial \mathcal{H}}{\partial S} = (1-\mathfrak{u}_1)\lambda_H^*(\lambda_1 - \lambda_6) + (1-\mathfrak{u}_2)\lambda_C^*(\lambda_1 - \lambda_5) + \mu\lambda_1,$$

$$\frac{d\lambda_2}{dt} = -\frac{\partial \mathcal{H}}{\partial C_q} = (1-\mathfrak{u}_1)\lambda_H^*(\lambda_2 - \lambda_6) + \alpha_1(\lambda_2 - \lambda_1) + \mu\lambda_2,$$

$$\frac{d\lambda_3}{dt} = -\frac{\partial \mathcal{H}}{\partial H_p} = (1-\mathfrak{u}_2)\lambda_C^*(\lambda_3 - \lambda_5) + \alpha_2(\lambda_3 - \lambda_1) + \mu\lambda_3,$$

$$\frac{d\lambda_4}{dt} = -\frac{\partial \mathcal{H}}{\partial C_v} = (1-\mathfrak{u}_1)\lambda_H^*(\lambda_4 - \lambda_6) + (1-\mathfrak{u}_2)\varepsilon\lambda_C^*(\lambda_4 - \lambda_5) + \rho(\lambda_4 - \lambda_1) + \mu\lambda_4,$$

$$\frac{d\lambda_5}{dt} = -\frac{\partial \mathcal{H}}{\partial C_i} = -\mathfrak{w}_1 + (1-\mathfrak{u}_2)\beta_2 A_1^*(\lambda_1 - \lambda_5) + (1-\mathfrak{u}_2)\beta_2 A_3^*(\lambda_3 - \lambda_5) + (1-\mathfrak{u}_2)\varepsilon\beta_2 A_4^*(\lambda_4 - \lambda_5) +$$
$$(1-\mathfrak{u}_2)\phi_1\beta_2 A_6^*(\lambda_6 - \lambda_8) + (1-\mathfrak{u}_2)\phi_2\beta_2 A_7^*(\lambda_7 - \lambda_9) + (1-\mathfrak{u}_1)\upsilon\lambda_H(\lambda_5 - \lambda_8) + (\mu + d_1)\lambda_5 + \mathfrak{u}_3\kappa(\lambda_5 - \lambda_{10}),$$

$$\frac{d\lambda_6}{dt} = -\frac{\partial \mathcal{H}}{\partial H_u} = (1-\mathfrak{u}_1)\frac{\beta_1}{N}A_1^*(\lambda_1 - \lambda_6) + (1-\mathfrak{u}_1)\frac{\beta_1}{N}A_2^*(\lambda_2 - \lambda_6) + (1-\mathfrak{u}_1)\frac{\beta_1}{N}A_4^*(\lambda_4 - \lambda_6) +$$
$$(1-\mathfrak{u}_1)\upsilon\frac{\beta_1}{N}A_5^*(\lambda_5 - \lambda_8) + (1-\mathfrak{u}_2)\phi_1\lambda_C^*(\lambda_6 - \lambda_8) + (\mu + d_2)\lambda_6 + \Theta(\lambda_6 - \lambda_7),$$

$$\frac{d\lambda_7}{dt} = -\frac{\partial \mathcal{H}}{\partial H_a} = -\mathfrak{w}_2 + (1-\mathfrak{u}_1)\frac{\beta_1\rho_1}{N}A_1^*(\lambda_1 - \lambda_6) + (1-\mathfrak{u}_1)\frac{\beta_1\rho_1}{N}A_2^*(\lambda_2 - \lambda_6) + (1-\mathfrak{u}_1)\frac{\beta_1\rho_1}{N}A_4^*(\lambda_4 - \lambda_6) +$$
$$(1-\mathfrak{u}_1)\upsilon\frac{\beta_1\rho_1}{N}A_5^*(\lambda_5 - \lambda_8) + (1-\mathfrak{u}_2)\phi_2\lambda_C^*(\lambda_7 - \lambda_9) + (d_3 + \mu)\lambda_7 + \mathfrak{u}_4\gamma(\lambda_7 - \lambda_{11}),$$

$$\frac{d\lambda_8}{dt} = -\frac{\partial \mathcal{H}}{\partial M_u} = -\mathfrak{w}_3 + (1-\mathfrak{u}_1)\frac{\beta_1\rho_2}{N}A_1^*(\lambda_1 - \lambda_6) + (1-\mathfrak{u}_2)\beta_2\omega_1 A_1^*(\lambda_1 - \lambda_5) + (1-\mathfrak{u}_1)\frac{\beta_1\rho_2}{N}A_2^*(\lambda_2 - \lambda_6) +$$
$$(1-\mathfrak{u}_2)\beta_2\omega_1 A_3^*(\lambda_3 - \lambda_5) + (1-\mathfrak{u}_1)\frac{\beta_1\rho_2}{N}A_4^*(\lambda_4 - \lambda_6) + (1-\mathfrak{u}_2)\varepsilon\beta_2\omega_1 A_4^*(\lambda_4 - \lambda_5) + (1-\mathfrak{u}_1)\upsilon\frac{\beta_1\rho_2}{N}A_5^*$$
$$(\lambda_5 - \lambda_8) + \mathfrak{u}_3\theta_1(\lambda_8 - \lambda_6) + (1-\mathfrak{u}_2)\phi_1\beta_2\omega_1 A_6^*(\lambda_6 - \lambda_8) + (1-\mathfrak{u}_2)\phi_2\beta_2\omega_1 A_7^*(\lambda_7 - \lambda_9) + (\mu + d_4)\lambda_8 + \delta(\lambda_8 - \lambda_9),$$

$$\frac{d\lambda_9}{dt} = -\frac{\partial \mathcal{H}}{\partial M_a} = -\mathfrak{w}_4 + (1-\mathfrak{u}_1)\frac{\beta_1\rho_3}{N}A_1^*(\lambda_1 - \lambda_6) + (1-\mathfrak{u}_2)\beta_2\omega_2 A_1^*(\lambda_1 - \lambda_5) + (1-\mathfrak{u}_1)\frac{\beta_1\rho_3}{N}A_2^*(\lambda_2 - \lambda_6) +$$
$$(1-\mathfrak{u}_2)\beta_2\omega_2 A_3^*(\lambda_3 - \lambda_5) + (1-\mathfrak{u}_1)\frac{\beta_1\rho_3}{N}A_4^*(\lambda_4 - \lambda_6) + (1-\mathfrak{u}_2)\varepsilon\beta_2\omega_2 A_4^*(\lambda_4 - \lambda_5) + (1-\mathfrak{u}_1)\upsilon\frac{\beta_1\rho_3}{N}A_5^*(\lambda_5 - \lambda_8) +$$
$$(1-\mathfrak{u}_2)\phi_1\beta_2\omega_2 A_6^*(\lambda_6 - \lambda_8) + \mathfrak{u}_3\theta_2(\lambda_9 - \lambda_7) + (1-\mathfrak{u}_2)\phi_2\beta_2\omega_2 A_7^*(\lambda_7 - \lambda_9) + (\mu + d_5)\lambda_9,$$

$$\frac{d\lambda_{10}}{dt} = -\frac{\partial \mathcal{H}}{\partial R} = -\eta\lambda_1 + (\mu + \eta)\lambda_{10},$$

$$\frac{d\lambda_{11}}{dt} = -\frac{\partial \mathcal{H}}{\partial H_t} = \mu\lambda_{11},$$

$$(26)$$

with transversality conditions

$$\lambda_i^*\left(T_f\right) = 0, \; i = 1, \; 2, \; \ldots, 11. \tag{27}$$

To obtain the control values, we compute the partial derivative of the Hamiltonian, given by:

$$\frac{\partial \mathcal{H}}{\partial \mathfrak{u}_i} = 0, \; \text{for } i = 1, \; 2, \; 3, \; 4 \tag{28}$$

Moreover, the corresponding optimal controls with the boundary condition of each control $\mathfrak{u}_1^*(t),\ \mathfrak{u}_2^*(t),\ \mathfrak{u}_3^*(t),\ $ and $\mathfrak{u}_4^*(t)$ are given by

$$\mathfrak{u}_1^*(t) = \max\left\{0,\ min\left\{\frac{\lambda_H^* S^*(\lambda_6 - \lambda_1) + \lambda_H^* C_q^*(\lambda_6 - \lambda_2) + \lambda_H^* C_v^*(\lambda_6 - \lambda_4) + \upsilon\lambda_H^* C_i^*(\lambda_8 - \lambda_5)}{\mathfrak{B}_1},\ 1\right\}\right\},$$

$$\mathfrak{u}_2^*(t) = \max\left\{0,\ min\left\{\frac{\lambda_C^* S^*(\lambda_5 - \lambda_1) + \lambda_C^* H_p^*(\lambda_5 - \lambda_3) + \varepsilon\lambda_C^* C_v^*(\lambda_5 - \lambda_4) + \phi_1\lambda_C^* H_u^*(\lambda_8 - \lambda_6) + \phi_2\lambda_C^* H_p^*(\lambda_9 - \lambda_7)}{\mathfrak{B}_2},\ 1\right\}\right\}$$

$$\mathfrak{u}_3^*(t) = \max\left\{0,\ min\left\{\frac{\Theta_1 M_u^*(\lambda_8 - \lambda_6) + \Theta_2 M_a^*(\lambda_9 - \lambda_7) + \kappa C_i^*(\lambda_5 - \lambda_{10})}{\mathfrak{B}_3},\ 1\right\}\right\},$$

$$\mathfrak{u}_4^*(t) = \max\left\{0,\ min\left\{\frac{\gamma H_p^*(\lambda_7 - \lambda_{11})}{\mathfrak{B}_4},\ 1\right\}\right\}. \tag{29}$$

From the previous analysis, to get the optimal point, we have to solve the system

$$\dot{S}^* = k_1\Delta + \alpha_1 C_q^* + \alpha_2 H_p^* + \rho C_v^* + \eta R^* - (1 - \mathfrak{u}_1^*)\lambda_H^* S^* - (1 - \mathfrak{u}_2^*)\lambda_C^* S^* - \mu S^*,$$

$$\dot{C}_q^* = k_2\Delta - \left((1 - \mathfrak{u}_1^*)\lambda_H^* + \alpha_1 + \mu\right)C_q^*,$$

$$\dot{H}_p^* = k_3\Delta - \left(\alpha_2 + \mu + (1 - \mathfrak{u}_2^*)\lambda_C^*\right)H_p^*,$$

$$\dot{C}_v^* = k_4\Delta - \left(\rho + \mu + (1 - \mathfrak{u}_1^*)\lambda_H^* + \varepsilon(1 - \mathfrak{u}_2^*)\lambda_C^*\right)C_v^*,$$

$$\dot{C}_i^* = (1 - \mathfrak{u}_2^*)\lambda_C^* S^* \lambda_C^* H_p^* + \varepsilon(1 - \mathfrak{u}_2^*)\lambda_C^* H_p^* - \upsilon(1 - \mathfrak{u}_1^*)\lambda_H^* C_i^* - (\mu + d_1 + \mathfrak{u}_3^*\kappa)C_i^*,$$

$$\dot{H}_u^* = (1 - \mathfrak{u}_1^*)\lambda_H^* S^* + (1 - \mathfrak{u}_1^*)\lambda_H^* C_q^* + (1 - \mathfrak{u}_1^*)\lambda_H^* C_v^* + \mathfrak{u}_3^*\theta_1 M_u^* - (1 - \mathfrak{u}_2^*)\phi_1\lambda_C^* H_u^* - (\theta + \mu + d_2)H_u^*,$$

$$\dot{H}_a^* = \Theta H_u^* + \mathfrak{u}_3^*\Theta_2 M_a^* - (1 - \mathfrak{u}_2^*)\phi_2\lambda_C^* H_a^* - (\mathfrak{u}_4^*\gamma + d_3 + \mu)H_a^*,$$

$$\dot{M}_u^* = (1 - \mathfrak{u}_2^*)\phi_1\lambda_C^* H_u^* + (1 - \mathfrak{u}_1^*)\upsilon\lambda_H^* C_i^* - (\mu + d_4 + \delta + \mathfrak{u}_3^*\Theta_1)M_u^*,$$

$$\dot{M}_a^* = \delta M_u^* + (1 - \mathfrak{u}_2^*)\phi_2\lambda_C^* H_a^* - (\mu + d_5 + \mathfrak{u}_3^*\theta_2)M_a^*,$$

$$\dot{R}^* = \mathfrak{u}_3^*\kappa C_i^* - (\mu + \eta)R^*,$$

$$\dot{H}_t^* = \mathfrak{u}_4^*\gamma A_7^* - \mu H_t^*,$$

with the Hamiltonian

$$\mathcal{H} = \mathfrak{w}_1 C_i^* + \mathfrak{w}_2 H_a^* + \mathfrak{w}_3 M_u^* + \mathfrak{w}_4 M_a^* + \frac{\mathfrak{B}_1}{2}\left(\mathfrak{u}_1^*\right)^2 + \frac{\mathfrak{B}_2}{2}\left(\mathfrak{u}_2^*\right)^2 + \frac{\mathfrak{B}_3}{2}\left(\mathfrak{u}_3^*\right)^2 + \frac{\mathfrak{B}_4}{2}\left(\mathfrak{u}_4^*\right)^2 + \lambda_1(k_1\Delta + \alpha_1 C_q^* + \alpha_2 H_p^* + \rho C_v^* + \eta R^* -$$

$$(1 - \mathfrak{u}_1^*)\lambda_H^* S^* - (1 - \mathfrak{u}_2^*)\lambda_C^* S^* - \mu S^*)$$

$$+ \lambda_2\left(k_2\Delta - \left((1 - \mathfrak{u}_1^*)\lambda_H^* + \alpha_1 + \mu\right)C_q^*\right) + \lambda_3(k_3\Delta - \left(\alpha_2 + \mu + (1 - \mathfrak{u}_2^*)\lambda_C^*\right)H_p^*) + \lambda_4(k_4\Delta - \left(\rho + \mu + (1 - \mathfrak{u}_1^*)\lambda_H^* + \varepsilon(1 - \mathfrak{u}_2^*)\lambda_C^*\right)C_v^*) +$$

$$\lambda_5((1 - \mathfrak{u}_2^*)\lambda_C^* S^* + (1 - \mathfrak{u}_2^*)\lambda_C^* H_p^* + \varepsilon(1 - \mathfrak{u}_2^*)\lambda_C^* H_p^* - \upsilon(1 - \mathfrak{u}_1^*)\lambda_H^* C_i^* - (\mu + d_1 + \mathfrak{u}_3^*\kappa)C_i^*) +$$

$$\lambda_6\left((1 - \mathfrak{u}_1^*)\lambda_H^* S^* + (1 - \mathfrak{u}_1^*)\lambda_H^* C_q^* + (1 - \mathfrak{u}_1^*)\lambda_H^* C_v^* + \mathfrak{u}_3^*\theta_1 M_u^* - (1 - \mathfrak{u}_2^*)\phi_1\lambda_C^* H_u^* - (\theta + \mu + d_2)H_u^*\right)$$

$$+ \lambda_7\left(\theta H_u^* + \mathfrak{u}_3^*\theta_2 M_a^* - (1 - \mathfrak{u}_2^*)\phi_2\lambda_C^* H_a^* - (\mathfrak{u}_4^*\gamma + d_3 + \mu)H_a^*\right) + \lambda_8\left((1 - \mathfrak{u}_2^*)\phi_1\lambda_C^* H_u^* + (1 - \mathfrak{u}_1^*)\upsilon\lambda_H^* H_u^* - (\mu + d_4 + \delta + \mathfrak{u}_3^*\theta_1)M_u^*\right)$$

$$+ \lambda_9\left(\delta M_u^* + (1 - \mathfrak{u}_2^*)\phi_2\lambda_C^* H_a^* - (\mu + d_5 + \mathfrak{u}_3^*\theta_2)M_a^*\right) + \lambda_{10}\left(\mathfrak{u}_3^*\kappa H_u^* - (\mu + \eta)R^*\right) + \lambda_{11}\left(\mathfrak{u}_4^*\gamma H_a^* - \mu H_t^*\right),$$

where

$$\lambda_H^* = \frac{\beta_1}{N}\left(H_u^*(t) + \rho_1 H_a^*(t) + \rho_2 M_u^*(t) + \rho_3 M_a^*(t)\right) \text{ and } \lambda_C^* = \beta_2\left(H_u^*(t) + \omega_1 M_u^*(t) + \omega_2 M_a^*(t)\right).$$

## 5. Numerical results

In this section we have presented the numerical result we have obtained using the parameters value collected in Table 3 below. We have collected data from a variety of sources, and have compiled the values in the table for the convenience of the constructed model numerical simulations and to verify the analytical results.

### 5.1. Numerical simulations and discussions of the deterministic model (3)

In this section, a numerical simulation of the entire HIV/AIDS and COVID-19 co-infection model given in Eq (3) is performed. We used ode45 fourth order Runge-Kutta scheme to examine the effect of various parameters on the spread and control of COVID-19 mono-infection, HIV/AIDS mono-infection, and HIV/AIDS and COVID-19 co-infection. The parameter

Table 3. Parameter values used for the co-infection model simulation.

| Symbol | Value | Source |
|---|---|---|
| $\Delta$ | 2500 | [32] |
| $\mu$ | 0.019 | [32] |
| $\alpha_1$ | 0.31 | Assumed |
| $\alpha_2$ | 0.43 | Estimated from [1] |
| $d_1$ | 0.33 | [39] |
| $d_2$ | 0.315 | [26] |
| $\theta$ | 0.21 | [43] |
| $\theta_1$ | 0.30 | [43] |
| $\theta_2$ | 0.30 | [43] |
| $\theta_3$ | 0.38 | Assumed |
| $\upsilon$ | 0.3 | [43] |
| $\phi_1$ | 1 | [43] |
| $\phi_2$ | 1 | [43] |
| $\rho_1$ | 1.25 | Assumed |
| $\rho_2$ | 1.5 | Assumed |
| $\rho_2$ | 1.8 | Assumed |
| $\gamma$ | 0.2 | [10] |
| $d_3$ | 0.34 | Assumed |
| $d_4$ | 0.42 | Assumed |
| $d_5$ | 0.51 | Assumed |
| $\eta$ | 0.200 | [26] |
| $\delta$ | 0.53 | Assumed |
| $\varepsilon$ | 0.002 | [14] |
| $\beta_1$ | 0.3425 | [43] |
| $\beta_2$ | 0.1175 | [43] |
| $\omega_1$ | 1.1 | Assumed |
| $\omega_2$ | 1.4 | Assumed |
| $k_1$ | 0.40 | [27] |
| $k_2$ | 0.20 | Assumed |
| $k_3$ | 0.20 | Assumed |
| $k_4$ | 0.20 | Assumed |
| $\rho$ | 0.30 | Assumed |
| $\kappa$ | 0.05 | [23] |
| $\upsilon$ | 0.85 | Assumed |

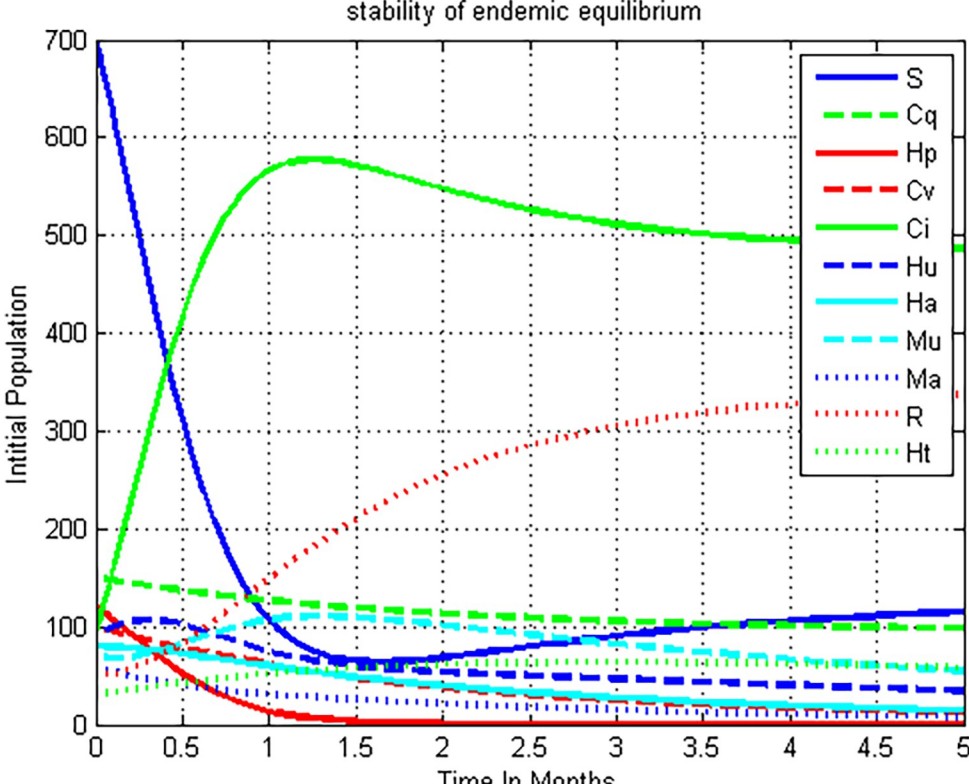

**Fig 3. The feature of the co-infection model (3) solutions at $\mathcal{R}_0^{HC} = 3.2 > 1$.**

values presented in Table 3 are used for numerical simulation. Moreover, we have investigated the stability of the endemic equilibrium point of the co-infection model (3), the effects of parameter on reproduction numbers, and the impact of treatment primarily on dually-infected individuals in the community.

## 5.2. Simulation of co-infection model (3) whenever $\mathcal{R}_0^{HC} = 3.2 > 1$

The above Fig 3 was plotted using ode45 Runge-Kutta fourth order method to observe the numerical simulation of the full co-infection model (3) by using parameter values from Table 3. We can deduce from the figure that after a year, the solutions of the COVID-19 and HIV/AIDS co-infection dynamical system (3) are approaching the endemic equilibrium point of the given dynamical system whenever the co-infection effective reproduction number $\mathcal{R}_0^{HC} = \max\{\mathcal{R}_{HM}, \mathcal{R}_{CM}\} = \max\{2, 3.2\} = 3.2 > 1$.

## 5.3. Numerical simulation to show the effect of $k_3$ on $\mathcal{R}_{HM}$

The effect of the HIV protection rate on the HIV/AIDS effective reproduction number $\mathcal{R}_{HM}$ is depicted in Fig 4. The graph shows that as the value of protection rate $k_3$ increases, the effective reproduction number $\mathcal{R}_{HM}$ decreases and for $k_3 > 0.771$ indicates that $\mathcal{R}_{HM}$ is reduced to less than one. As a result, the public health and policymakers must focus on increasing the values of the HIV/AIDS protection rate $k_3$ in order to control HIV/AIDS spread which may causes for existence of co-infection in the community.

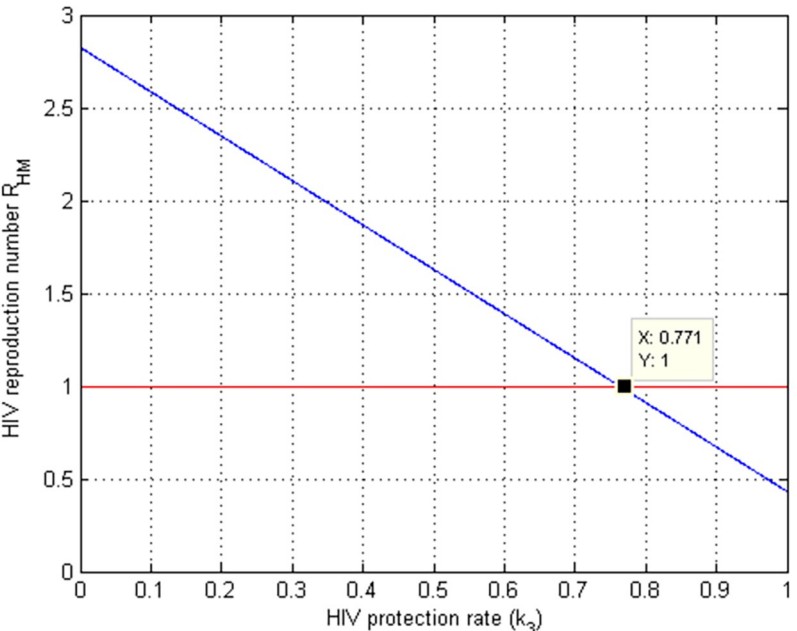

**Fig 4. Simulation of HIV protection rate k3 versus $\mathcal{R}_{HM}$.**

## 5.4. Simulation to show the effect of $\kappa$ on $\mathcal{R}_C$

A numerical simulation in order to show the effect of COVID-19 treatment on the COVID-19 effective reproduction number $\mathcal{R}_C$ is given by Fig 5. The graph shows that as the value of the treatment rate raises, the COVID-19 basic reproduction number decreases and for the value of $\kappa > 0.776$ implies that $\mathcal{R}_C < 1$.

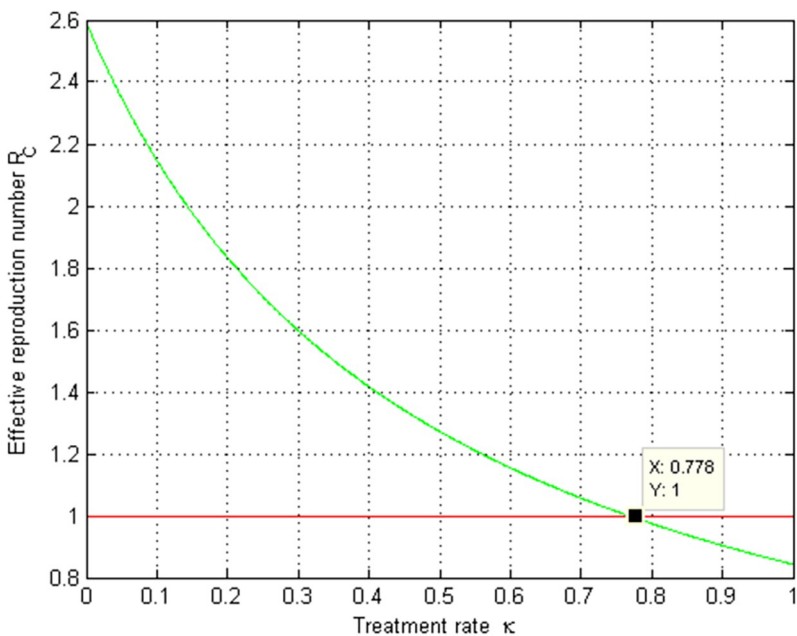

**Fig 5. Simulation of COVID-19 treatment rate κ versus $\mathcal{R}_C$.**

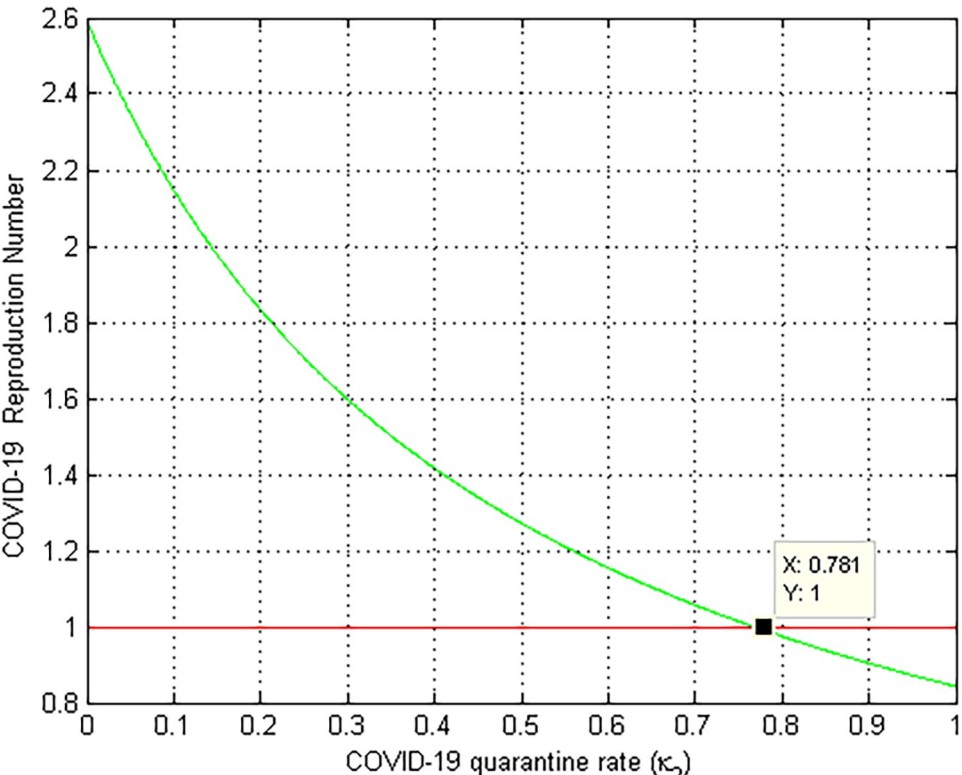

**Fig 6. Numerical simulation of COVID-19 protection rate $k_2$ versus $\mathcal{R}_{CM}$.**

## 5.5. Numerical simulation to show the effect of $k_2$ on $\mathcal{R}_{CM}$

Fig 6 depicted the effect of the COVID-19 protection rate $k_2$ on the COVID-19 effective reproduction number $\mathcal{R}_C$. As we can observe from the graph as the value of $k_2$ increases, the COVID-19 effective reproduction number decreases, and $k_2 > 0.654$ implies that $\mathcal{R}_C < 1$. As a result, all the stakeholders must focus on increasing the values of COVID-19 quarantine rate $k_2$ in order to prevent and control COVID-19 spread in the community. Biologically, this means that COVID-19 infection decreases as the quarantine rate $k_2$ rises.

## 5.6. Numerical simulation to show effect of $\beta_2$ on $\mathcal{R}_C$

Fig 7 shows the influence of the COVID-19 transmission rate $\beta_2$ on the COVID-19 effective reproduction number $\mathcal{R}_C$. The graph shows that as the value of $\beta_2$ rises, so does the COVID-19 effective reproduction number and the value of $\beta_2 < 0.225$ means that $\mathcal{R}_C < 1$. As a result, public health authorities must focus on reducing the value of COVID-19 transmission rate $\beta_2$ in order to avoid and regulate COVID-19 spread in the community.

## 5.7. Simulation to show effect of $\beta_1$ on $\mathcal{R}_{HM}$

Fig 8 depicts a numerical simulation on the influence of HIV transmission rate $\beta_1$ on the HIV/AIDS effective reproduction number $\mathcal{R}_{HM}$. The graph shows that as the value of $\beta_1$ grows, so does the HIV/AIDS effective reproduction number and whenever $\beta_1 < 0.193$ significantly $\mathcal{R}_{HM}$ reduces to less than unity. Therefore it is recommendable to give an attention on minimizing the value of the HIV transmission rate $\beta_1$ to prevent and control HIV/AIDs expansion

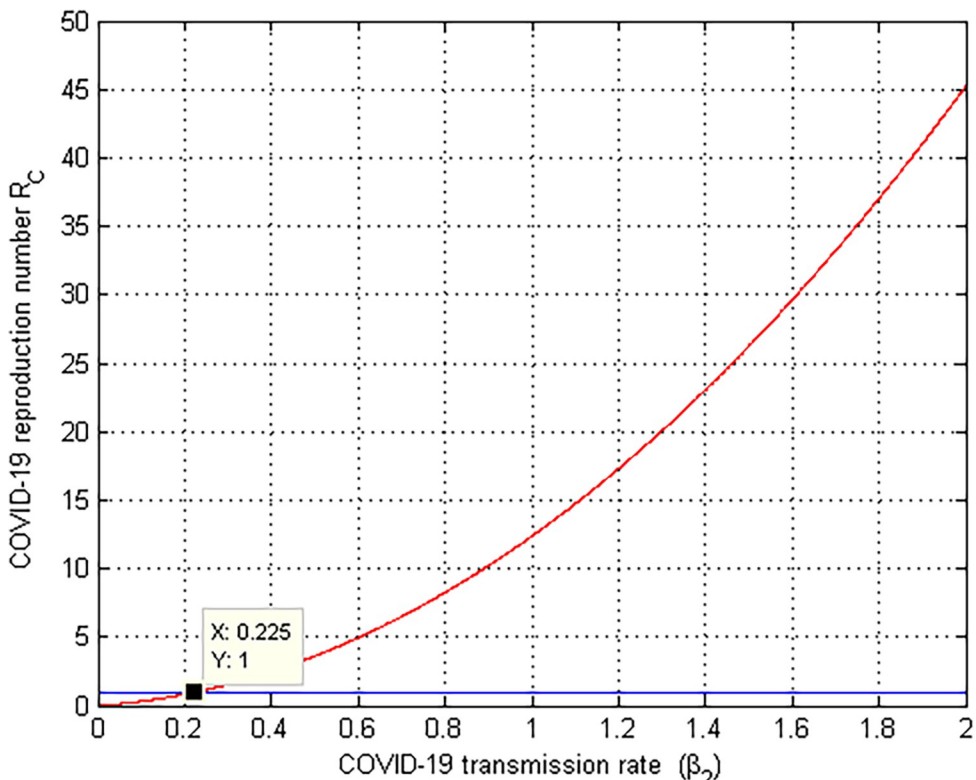

**Fig 7. Numerical simulation on $\beta_2$ versus $\mathcal{R}_C$.**

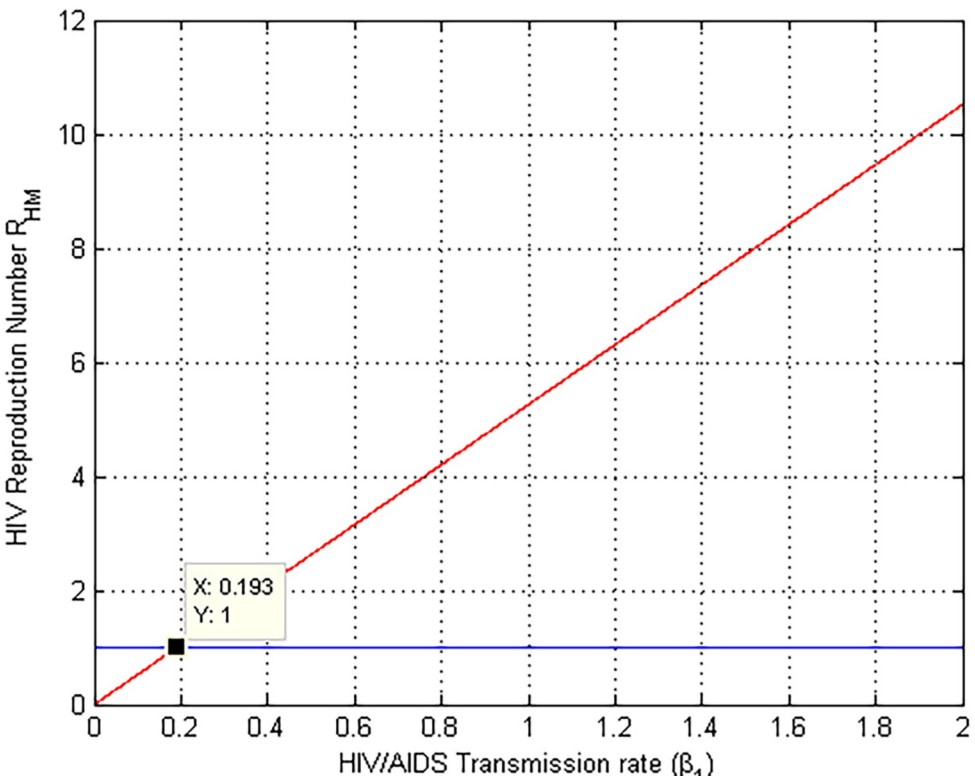

**Fig 8. Numerical simulation on $\beta_1$ versus $\mathcal{R}_{HM}$.**

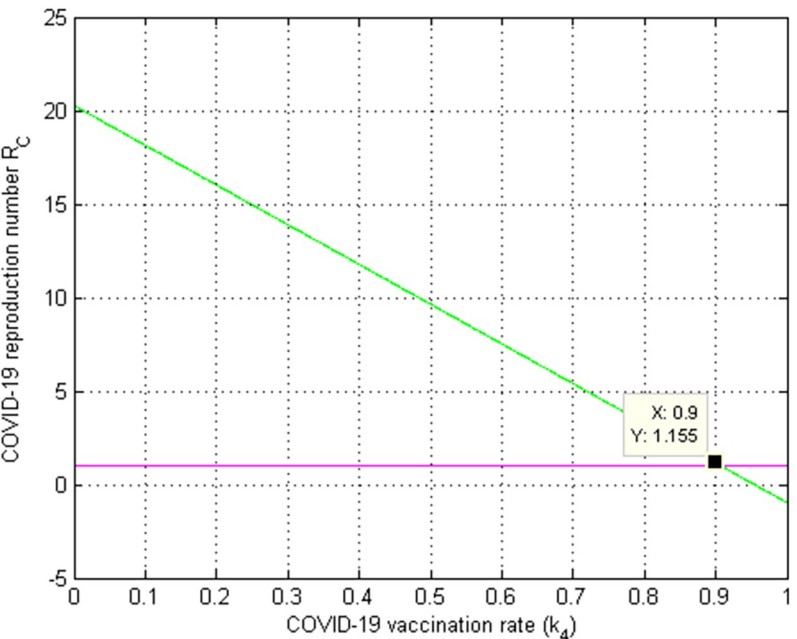

**Fig 9. Numerical simulation on $k_4$ versus $\mathcal{R}_C$.**

in the community. Biologically, this indicates that the HIV/AIDS infection lowers as the transmission rate $\beta_1$ drops.

## 5.8. Simulation to show effect of $k_4$ on $\mathcal{R}_C$

Fig 9 looked at how the COVID-19 immunization (vaccination) rate $k_4$ affected the COVID-19 effective reproduction number $\mathcal{R}_C$. The graph shows that when the value of $k_4$ grows, the COVID-19 effective reproduction number decreases, and values of $k_4 > 0.9$ suggest that $\mathcal{R}_C < 1$. As a result, public health authorities must focus on increasing the COVID-19 immunization rate $k_4$ in order to prevent and control COVID-19 spread in the community. Biologically, this indicates that the COVID-19 infection reduces as the immunization rate $k_4$ rises.

## 5.9. Numerical simulation to show effect of $\kappa$ on COVID-19 infectious ($C_i$)

Fig 10 examined the effect of COVID-19 treatment rate on the number of COVID-19 mono-infectious population. The graph shows that when the value of $\kappa$ increases, the number of COVID-19 mono-infectious people decrease. As a result, public officials should focus on increasing the value of the treatment rate at which COVID-19 infected individuals recovered from COVID-19 illness increase.

## 5.10. Simulation to show effect of $\theta_1$ on the co-infectious ($M_u$)

Fig 11 looked at how $\theta_1$ affected the number of COVID-19 and HIV/AIDS co-infected individuals. The graph shows that when the value of COVID-19 treatment rate $\theta_1$ rises, the number of COVID-19 and HIV/AIDS co-infected individuals' decreases. As a result, public officials should focus on maximizing the value of COVID-19 treatment rate $\theta_1$ in COVID-19 infected persons.

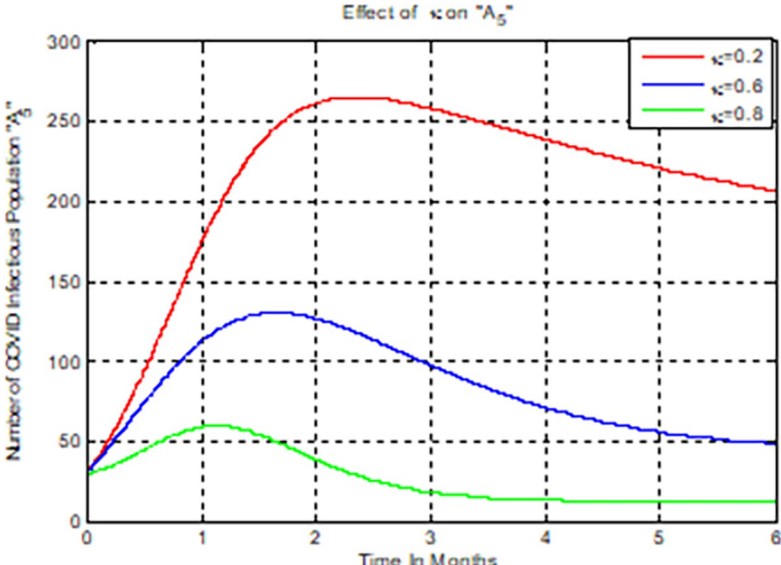

**Fig 10. Numerical simulations of $\kappa$ versus $C_i$.**

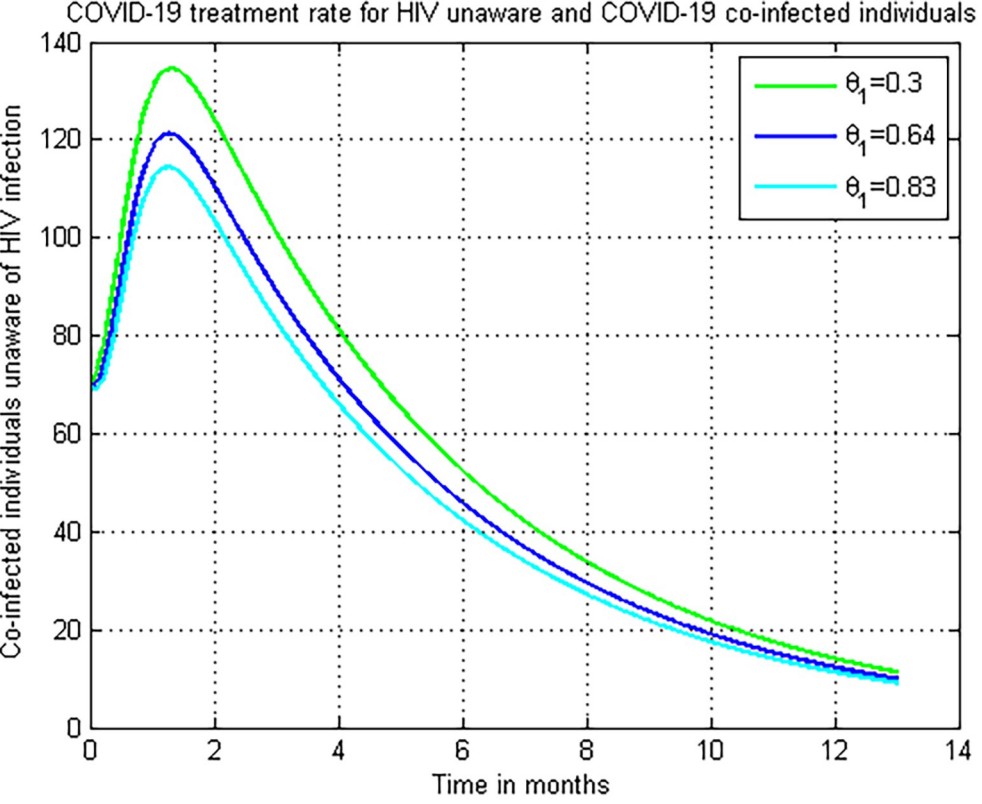

**Fig 11. Numerical simulation on $\theta_1$ versus $H_u$.**

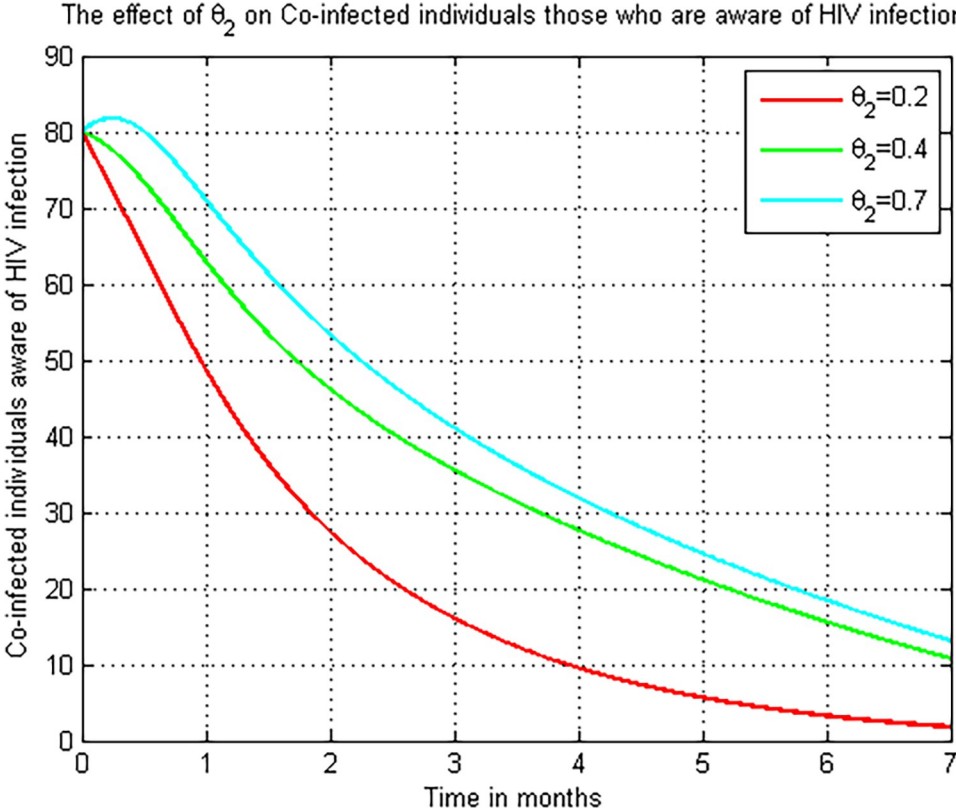

**Fig 12. Numerical simulation on $\theta_2$ versus $M_a$.**

## 5.11. Simulation to show effect of $\theta_2$ on the co-infectious ($M_a$)

Fig 12 show that the impact of $\theta_2$ on the number of COVID-19 and HIV/AIDS co-infected people. The graph shows that when the value of the COVID-19 treatment rate $\theta_2$ rises, the number of COVID-19 and HIV/AIDS co-infected individuals decrease. As a result, public officials must focus on maximizing the value of COVID-19 treatment rate $\theta_2$ in COVID-19 infected persons.

## 5.12. Numerical simulations of optimal control strategies

To verify the analytical results, the optimal control model system (17) is simulated using the parameter values given in Table 3 with positive weight constants $w_1 = w_2 = w_3 = w_4 = 18$. The optimal control system is composed of two dynamical systems, the state dynamical system (17) and the adjoint dynamical system (27), each with its own initial and final-time conditions, with the control value state in Eq (26). The fourth forward-backward Runge-Kutta iterative method is used to solve this optimality system. The state Eq (17) is solved with the initial values of state variables using the fourth-order forward Runge-Kutta method. We used backward fourth order Runge-Kutta to solve the adjoint equations once we had the solution of the state functions and the value of optimal controls. To determine the impact of control measures on the reduction of the HIV/AIDS and COVID-19 co-infection we have the following three cases of optimal control strategies:

**Case 1**: Controlling HIV infection $H_a$ with the combinations of strategies: strategy 1: use $u_1 = 0$, and $u_4 \neq 0$, strategy 2: use $u_1 \neq 0$, and $u_4 = 0$ and strategy 3: use $u_1 \neq 0$ and $u_4 \neq 0$.

**Case 2**: Controlling COVID-19 infection $C_i$ with the combinations of strategies: strategy 1: use $u_2 = 0$, and $u_3 \neq 0$, strategy 2: use $u_2 \neq 0$, and $u_3 = 0$ and strategy 3: use $u_2 \neq 0$, and $u_3 \neq 0$.

**Case 3**: Controlling the total HIV/AIDS and COVID-19 co-infection $M_u + M_a$ with the combinations of strategy 1: use the strategy $u_1 = 0$, $u_2 \neq 0$, $u_3 \neq 0$, and $u_4 \neq 0$ strategy 2: use the strategy $u_1 \neq 0$, $u_2 = 0$, $u_3 \neq 0$, and $u_4 \neq 0$ strategy 3: use the strategy $u_1 \neq 0$, $u_2 \neq 0$, $u_3 \neq 0$, and $u_4 \neq 0$ strategy 4: use the strategy $u_1 \neq 0$,

## 5.13. HIV infection ($H_a$) simulation with strategy 1 ($u_1 = 0$, and $u_4 \neq 0$)

In this subsection simulation is done for the HIV/AIDS infection ($H_a$) when there is no control strategy in place and when there is only HIV/AIDS treatment control measure. Fig 13 shows that the HIV/AIDS treatment control measure efforts are implemented then the number of individuals infected with HIV decreases throughout time to zero.

## 5.14. HIV infection simulation with strategy 1 ($u_1 \neq 0$, and $u_4 = 0$)

In this subsection simulation is done for the HIV/AIDS infection ($H_a$) when there is no control strategy in place and when there is only HIV/AIDS protection control measure. Fig 14 shows that the HIV/AIDS protection control measure efforts are implemented then the number of individuals infected with HIV decreases throughout time to zero.

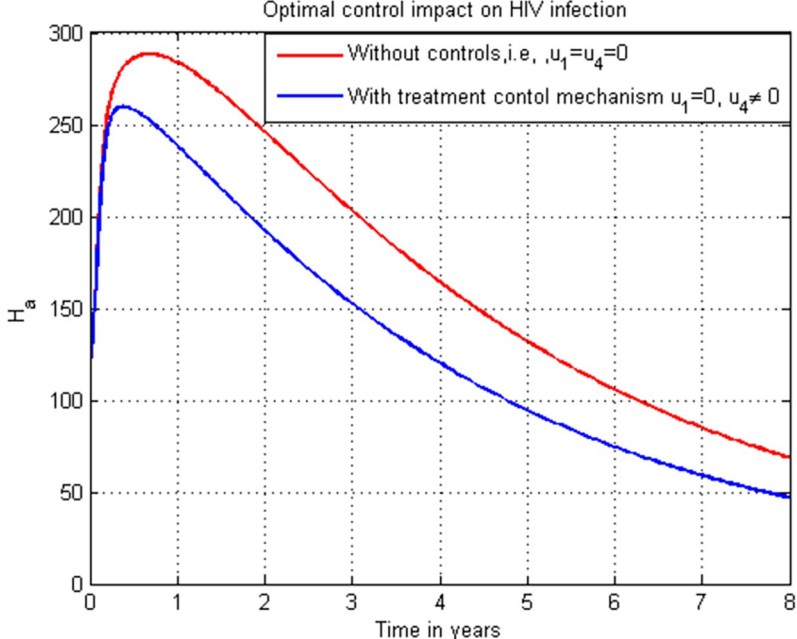

**Fig 13. Simulation of HIV infection ($H_a$) with HIV/AIDS treatment strategy.**

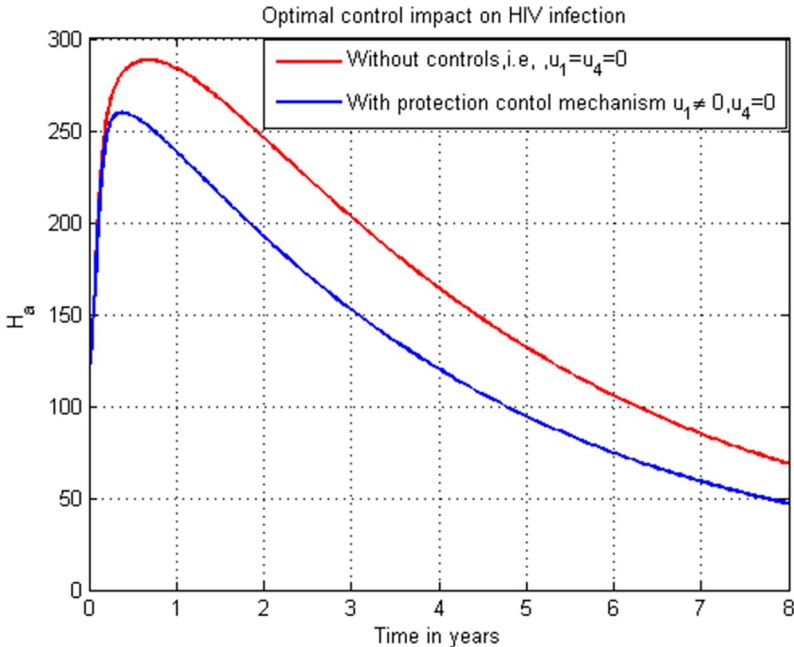

**Fig 14. Simulation of HIV infection ($H_a$) with HIV/AIDS protection strategy.**

### 5.15. HIV infection simulation with strategy 1 ($u_1 \neq 0$, and $u_4 \neq 0$)

In this subsection simulation is done for the HIV/AIDS infection ($H_a$) when there is no control strategy in place and when there are HIV/AIDS protection and treatment control measures. Fig 15 shows that the HIV/AIDS protection and treatment control measures efforts are

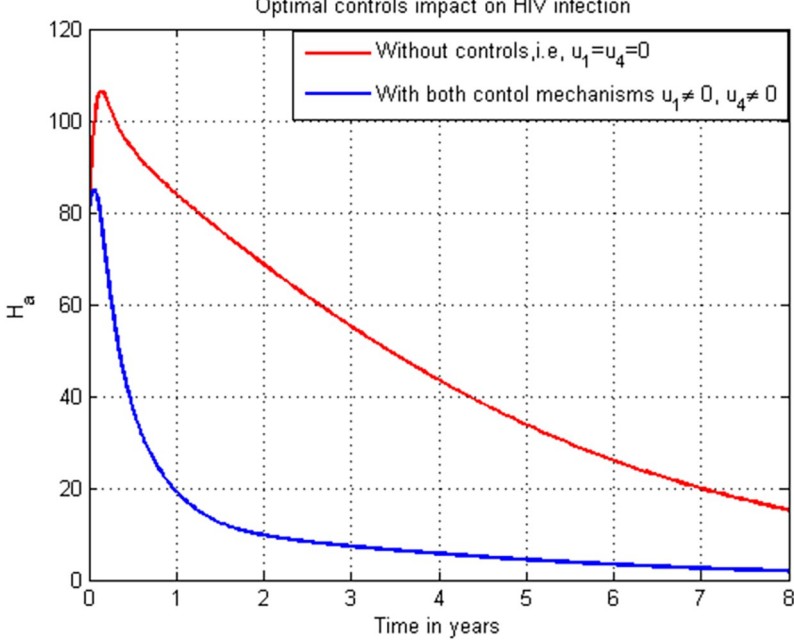

**Fig 15. Simulation of HIV infection ($H_a$) with both HIV/AIDS protection and treatment strategies.**

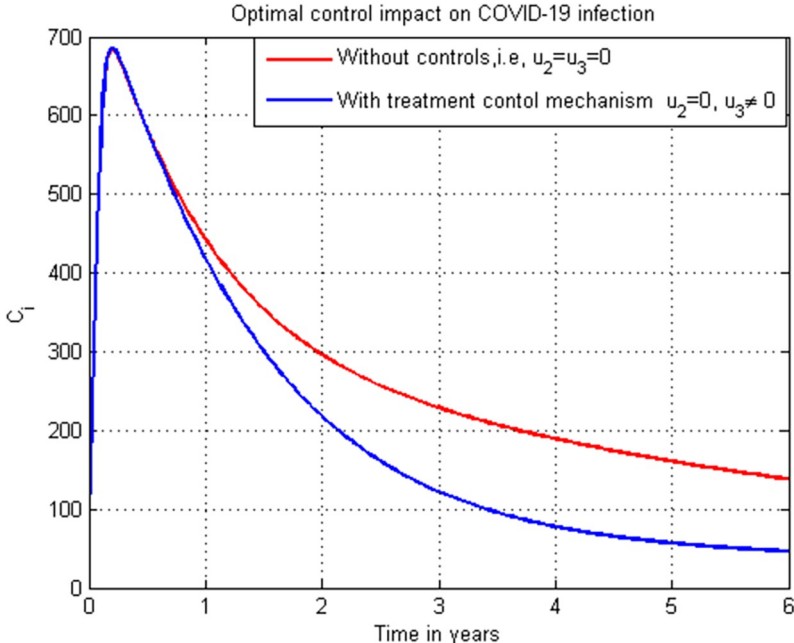

**Fig 16. Simulation of COVID-19 infection ($C_i$) with treatment strategy.**

implemented then the number of individuals infected with HIV/AIDS decreases rapidly to zero after seven years.

### 5.16. COVID-19 infection simulation with strategy 1 ($u_2 = 0$, and $u_3 \neq 0$)

In this subsection simulation is done for the COVID-19 infection ($C_i$) when there is no control strategy in place and when there is COVID-19 treatment control measure. Fig 16 shows that the COVID-19 treatment control measure effort is implemented then the number of individuals infected with COVID-19 decreases to zero through time.

### 5.17. COVID-19 infection simulation with strategy 1 ($u_2 \neq 0$, and $u_3 = 0$)

In this subsection simulation is done for the COVID-19 infection ($C_i$) when there is no control strategy in place and when there is COVID-19 protection control measure. Fig 17 shows that the COVID-19 protection control measure effort is implemented then the number of individuals infected with COVID-19 decreases to zero after five years.

### 5.18. COVID-19 infection simulation with strategy 1 ($u_2 \neq 0$, and $u_3 \neq 0$)

In this subsection simulation is done for the COVID-19 infection ($C_i$) when there is no control strategy in place and when there are COVID-19 protection and treatment control measures. Fig 18 shows that the COVID-19 protection and treatment control measures efforts are implemented then the number of individuals infected with COVID-19 decreases quickly to zero.

### 5.19. Co-infection simulation with strategy 1 ($u_1 = 0$, $u_2 \neq 0$, $u_3 \neq 0$, and $u_4 \neq 0$)

In this subsection simulation is done for the cumulated HIV/AIDS and COVID-19 co-infection when there is no control strategy in place and when there are controls involving COVID-

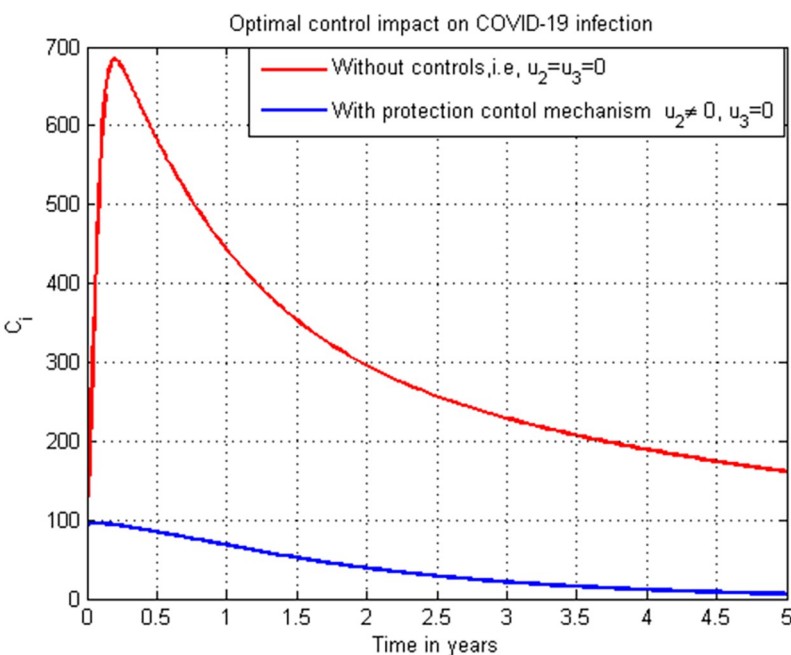

**Fig 17. Simulation of COVID-19 infection ($C_i$) with treatment strategy.**

19 protection, treatments for both HIV and COVID-19 single infections without HIV protection measure. Fig 19 shows the result that all the prevention and control strategies except HIV protection efforts are implemented, the number of individuals co-infected with HIV and COVID-19 decreases drastically to zero after year seven.

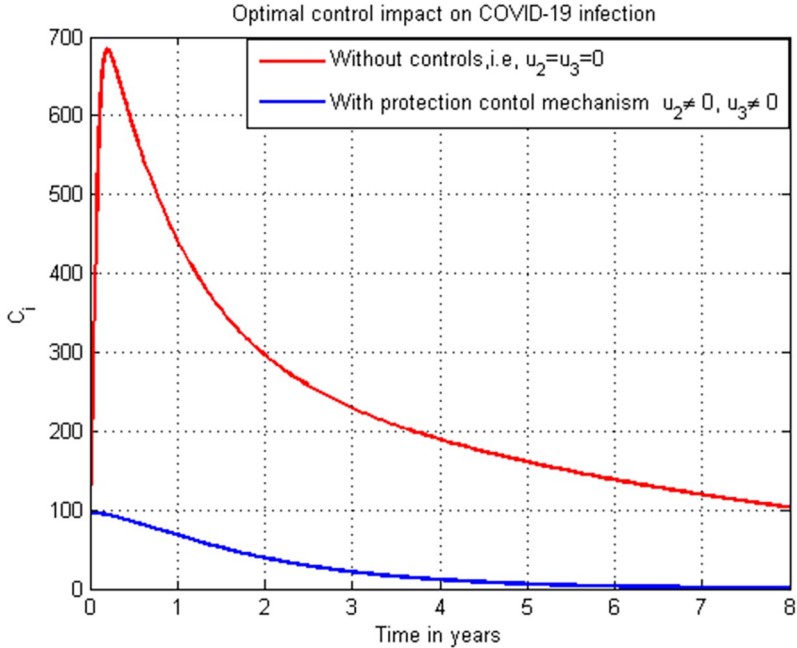

**Fig 18. Simulation of COVID-19 infection ($C_i$) with both protection and treatment strategies.**

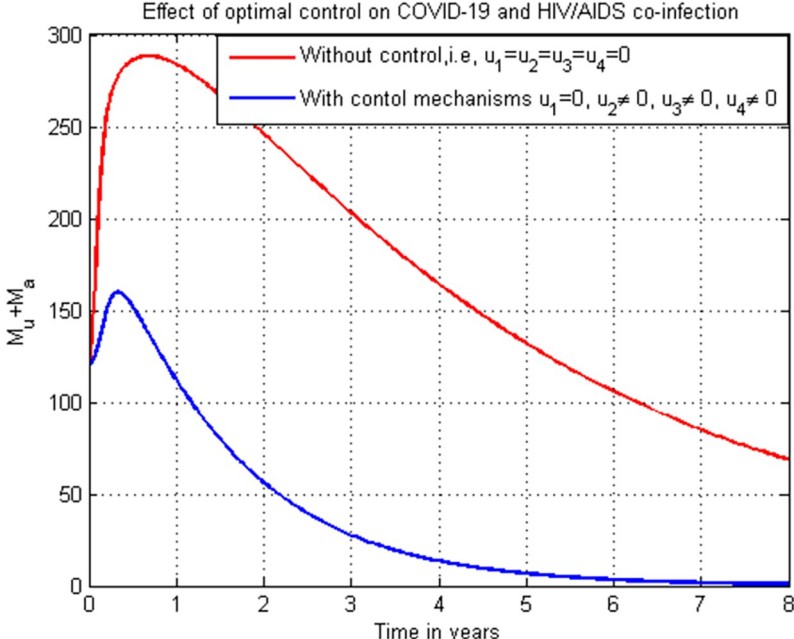

**Fig 19. Simulation of the co-infection with COVID-19 protection and COVID-19 and HIV/AIDS treatments strategies.**

## 5.20. Co-infection simulation with strategy 2 ($u_1 \neq 0$, $u_2 = 0$, $u_3 \neq 0$, and $u_4 \neq 0$)

In this subsection simulation is done when there is no control strategy in place and when there are controls involving HIV protection, treatment strategies for both HIV and COVID-19 single infections without COVID-19 protection measure. Fig 20 shows the result that all the prevention and control strategies except COVID-19 protection efforts are implemented, the number of individuals co-infected with HIV and COVID-19 decreases drastically to zero.

## 5.21. Co-infection simulation with strategy 3 ($u_1 \neq 0$, $u_2 \neq 0$, $u_3 = 0$, and $u_4 \neq 0$)

In this subsection simulation is done when there is no control strategy in place and when there are controls involving HIV protection, COVID-19 protection, and HIV treatment without COVID-19 treatment measure. Fig 21 shows the result that all the prevention and control strategies except HIV treatment strategy efforts are implemented, the number of individuals co-infected with HIV and COVID-19 decreases drastically to zero after 7 years.

## 5.22. Co-infection simulation with 4 ($u_1 \neq 0$, $u_2 \neq 0$, $u_3 \neq 0$, and $u_4 = 0$)

In this subsection simulation is done when there is no control strategy in place and when there are controls involving HIV protection, COVID-19 protection, and COVID-19 treatment without HIV treatment measures. Fig 22 shows the result that all the prevention and control strategies except HIV treatment strategy efforts are implemented, the number of individuals co-infected with HIV and COVID-19 decreases drastically to zero after 8 years.

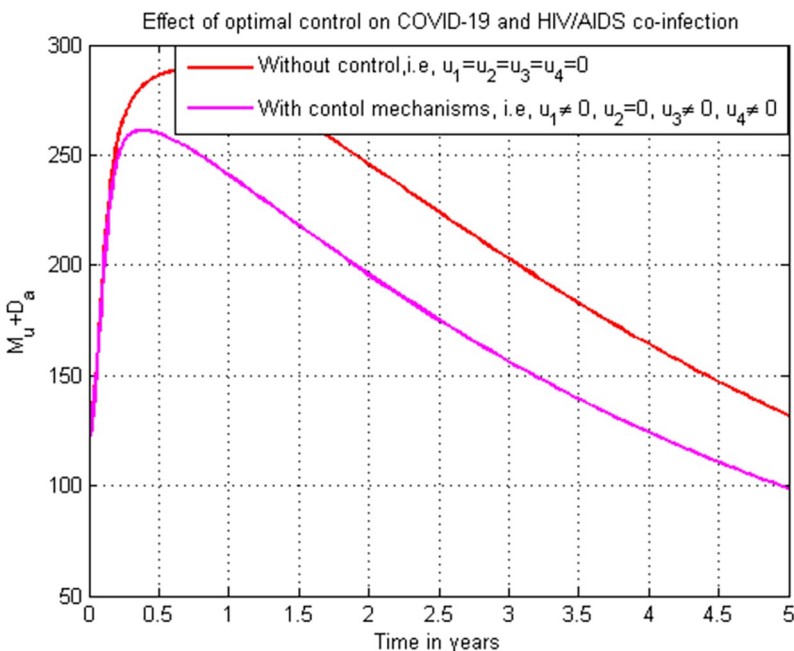

**Fig 20. Simulation of the co-infection with HIV protection and COVID-19 and HIV/AIDS treatments strategies.**

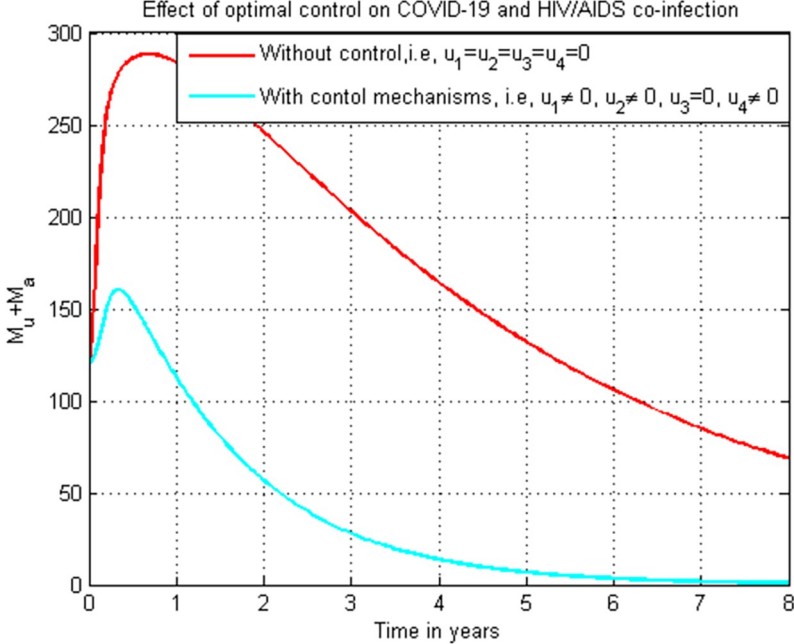

**Fig 21. Simulation of the co-infection with HIV protection, COVID-19 protection and HIV/AIDS treatment strategies.**

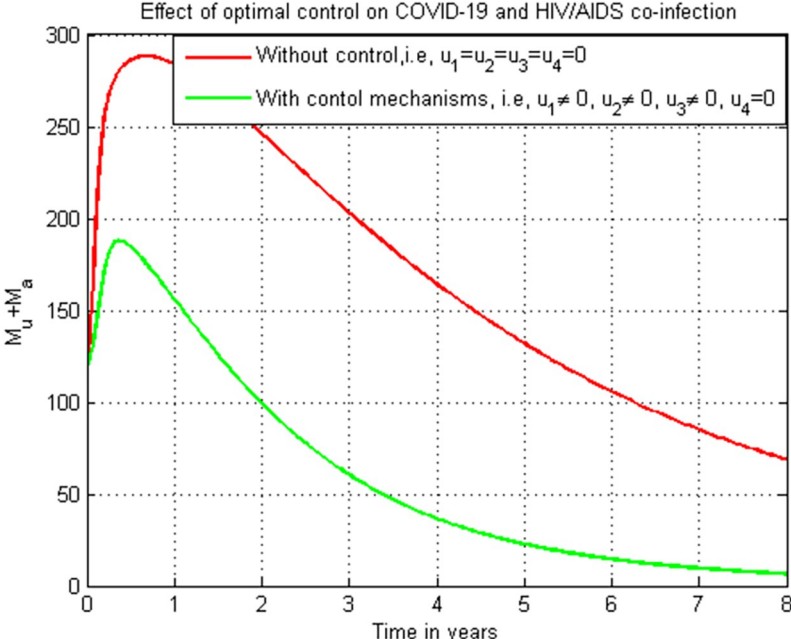

**Fig 22. Simulation of the co-infection with HIV protection, COVID-19 protection and COVID-19 treatment strategies.**

### 5.23. Co-infection simulation with strategy 5 ($u_1 = 0$, $u_2 = 0$, $u_3 \neq 0$, and $u_4 \neq 0$)

In this subsection simulation is done when there is no control strategy in place and when there are controls involving treatment strategies for COVID-19 and HIV single infection without HIV and COVID-19 protection measures. Fig 23 shows the result that treatment strategies efforts are implemented without protection strategies, the number of individuals co-infected with HIV and COVID-19 decreases drastically to zero in the long run.

### 5.24. Co-infection simulation with strategy 6 ($u_1 \neq 0$, $u_2 \neq 0$, $u_3 = 0$, and $u_4 = 0$)

In this subsection simulation is done when there is no control strategy in place and when there are control strategies involving protection strategies for COVID-19 and HIV single infection without HIV and COVID-19 treatment measures. Fig 24 shows the result that protection strategies efforts are implemented without treatment strategies, the number of individuals co-infected with HIV and COVID-19 decreases drastically to zero after 8 years later.

### 5.25. Co-infection simulation with strategy 7 ($u_1 \neq 0$, $u_2 \neq 0$, $u_3 \neq 0$, and $u_4 \neq 0$)

In this subsection simulation is done when there is no control strategy in place and when there are all the control strategies involving protection and treatment for both COVID-19 and HIV single infections. Fig 25 shows the result that all the protection and treatment strategies efforts are implemented, the number of individuals co-infected with HIV and COVID-19 decreases drastically to zero after 3.

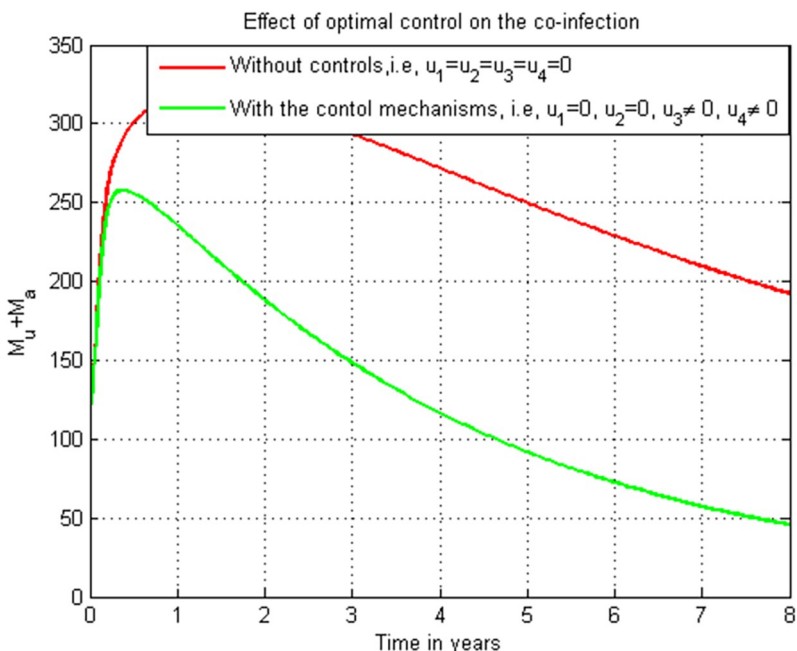

**Fig 23. Simulation of the co-infection with only HIV and COVID-19 treatments strategies.**

## 6. Conclusions

In this paper, we formulated and investigated a continuous time dynamical model for the transmission of HIV/AIDS and COVID-19 co-infection with protection and treatment strategies. The mode incorporate four non-infectious groups the susceptible group, the HIV

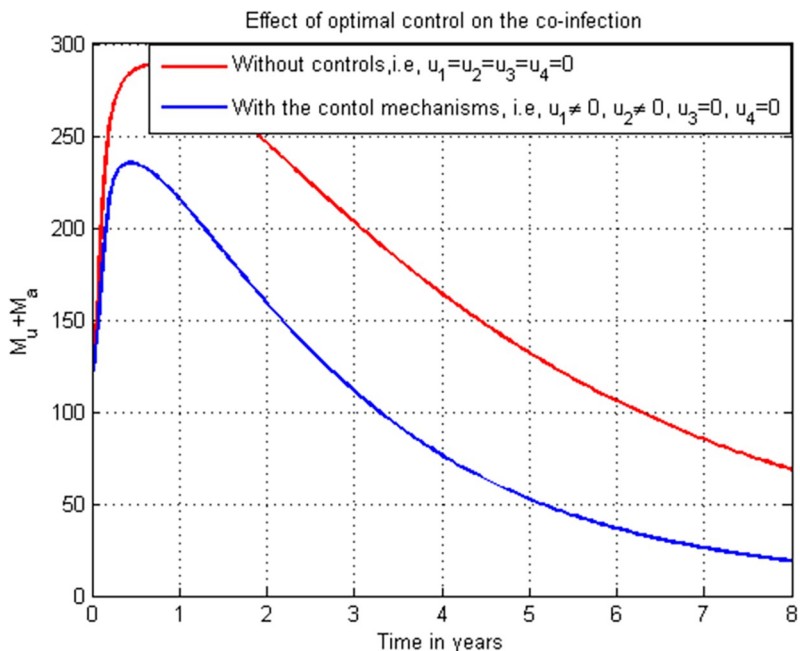

**Fig 24. Simulation of the co-infection with only HIV and COVID-19 protections strategies.**

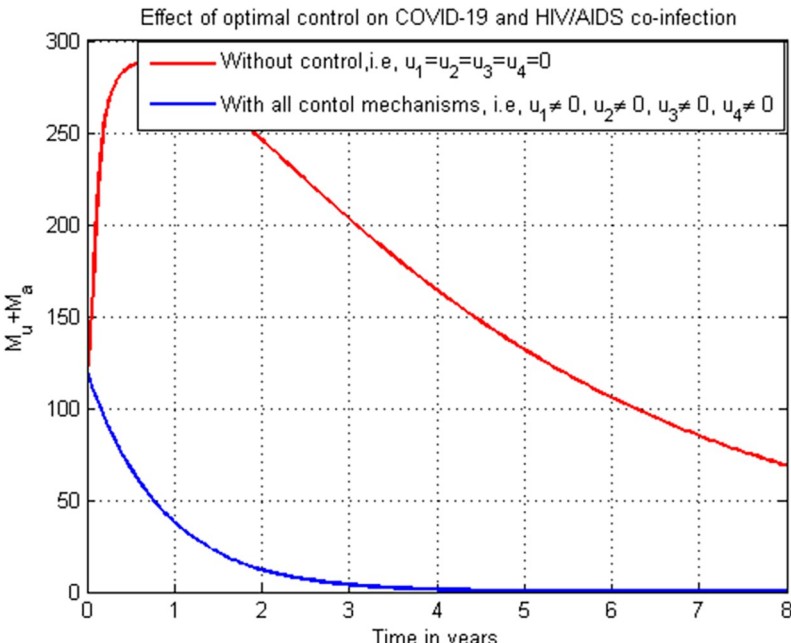

**Fig 25. Simulation of the co-infection with all possible strategies.**

protection group, the COVID-19 protection group, and the COVID-19 vaccinated group and this made the model highly nonlinear and challenging for the qualitative analysis of the co-infection model. The model has been mathematically analyzed both for the sub-models associating the cases that each disease type is isolated and in the case when there is co-infection. In addition an optimal control problem model that minimizes the cost of the infection as well as minimizes the control efforts to control the diseases transmission in the community is formulated and analyzed. The model includes the intervention strategies, protective as well as treatment and numerical simulations of both the deterministic model and optimal control problem models are presented. In the analysis it has been indicated that the effect of protection as well as treating the infected ones with the available treatment mechanisms affects significantly the optimal control strategy and its outcome. From the optimal control problem simulation results it can be concluded that applying both protective and treatment control mechanisms at the population level yields both economic as well as epidemiologic gains. Therefore, we recommended to the stake holders to give more attention and the overall optimal effort to implement both the protective as well as treatment control strategies to minimize the single infections as well as the co-infection diseases transmission in the community.

This study did not considered the stochastic method, fractional order method, impacts of the environment, structure of human age, the spatial structure, and real population primary epidemiological data. Based on these limitations potential researcher can consider to extend this study.

## Acknowledgments

We would like to give credit to Mr.Stotaw Ehete and Adugna Safeyi for their personal Wi-Fi contribution.

## Author Contributions

**Conceptualization:** Shewafera Wondimagegnhu Teklu.

**Data curation:** Belela Samuel Kotola.

**Formal analysis:** Shewafera Wondimagegnhu Teklu, Yohannes Fissha Abebaw.

**Investigation:** Belela Samuel Kotola.

**Methodology:** Belela Samuel Kotola.

**Software:** Belela Samuel Kotola, Yohannes Fissha Abebaw.

**Validation:** Belela Samuel Kotola.

**Visualization:** Belela Samuel Kotola, Yohannes Fissha Abebaw.

**Writing – original draft:** Belela Samuel Kotola.

**Writing – review & editing:** Belela Samuel Kotola.

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
