## [Decision Letter · Decision Letter 0]

3 Feb 2023

PONE-D-23-01945Bifurcation Analysis and Optimal Control for COVID-19 and HIV/AIDS Co-InfectionPLOS ONE

Dear Dr. Kotola,

Thank you for submitting your manuscript to PLOS ONE. After careful consideration, we feel that it has merit but does not fully meet PLOS ONE’s publication criteria as it currently stands. Therefore, we invite you to submit a revised version of the manuscript that addresses the points raised during the review process.

We look forward to receiving your revised manuscript.

Kind regards,

Oluwole Daniel Makinde, PhD

Academic Editor

PLOS ONE

Journal Requirements:

4. Please amend the manuscript submission data (via Edit Submission) to include authors Dr. Shewafera Wondimagegnhu Teklu and Dr. Yohannes Fissha Abebaw.

Additional Editor Comments:

See the reviewers improvement comments and revise the manuscript accordingly.

Reviewers' comments:

Reviewer's Responses to Questions

**Comments to the Author**

1. Is the manuscript technically sound, and do the data support the conclusions?

Reviewer #1: Yes

Reviewer #2: Partly

2. Has the statistical analysis been performed appropriately and rigorously? 

Reviewer #1: Yes

Reviewer #2: N/A

3. Have the authors made all data underlying the findings in their manuscript fully available?

Reviewer #1: Yes

Reviewer #2: Yes

4. Is the manuscript presented in an intelligible fashion and written in standard English?

Reviewer #1: Yes

Reviewer #2: Yes

5. Review Comments to the Author

Reviewer #1: Indeed, the paper is well written and organized. After checking the paper, I found it suitable for publication in this journal with some modifications. So that the authors should rephrase the article again. See the attached improvement comment.

Reviewer #2: Review Report

Title: Bifurcation Analysis and Optimal Control for COVID-19 and HIV/AIDS Co-Infection

The authors developed a mathematical model with 11 non-compatible compartments to study the transmission dynamics of both COVID-19 and HIV/AIDS. This compartmental model is a significant contribution to the field as it provides a more detailed understanding of the transmission dynamics of COVID-19 and HIV/AIDS in coexistence. The authors' use of 11 non-compatible classes enhances the accuracy and depth of the analysis, offering new insights into the complex relationships between these two diseases.

See attached file for more details

6. PLOS authors have the option to publish the peer review history of their article (what does this mean?). If published, this will include your full peer review and any attached files.

Reviewer #1: No

Reviewer #2: **Yes: **Alfred Hugo

---

## [Author Response · Author response to Decision Letter 0]

16 Feb 2023

Review Report 1

 The author could not present the main finding of the study. The failure to present the study’s key results detracts from the overall impact and relevance of the research. To make the study more impactful, the author should emphasize the main findings and highlight their significance in the context of the field.

Reflection: In the revised manuscript we have incorporated all the concepts mentioned. 

 Introduction 

 The manuscript lacks sufficient detail regarding co-infection. Information regarding the effects of simultaneous infection with COVID-19 and HIV/AIDS, the body’s physiological responses, and the underlying chemical processes is missing. 

Reflection: As per the request of the reviewer comments in the revised manuscript paragraphs 4 and 5 we have discussed the concept of co-infections and reviewed HIV/AIDS and COVID-19 literatures.

 In the second from last paragraph the first sentence starts with “Theoretically, they looked at the single infections infection-free equilibrium points’ local asymptotically stability if their estimated basic reproduction numbers were less than unity.” It seems to be not clear; what do you mean by this sentence 

Reflection: we strongly agree with the comment and we have removed the whole paragraph and replaced with relevant paragraph to this study.

 The last paragraph needs more evidence and how sure are that “Despite the fact that many researchers devote significant time and effort to researching HIV/AIDS, none of them has considered COVID-19 protection, COVID-19 vaccination, COVID-19 quarantine with treatment, HIV protection, and HIV treatment as prevention and control strategies in their model formulation. Furthermore, no other academics have conducted this research, and the abovementioned literature has encouraged and empowered us to do this research and pick up the slack” 

Reflection: We have restructured the paragraph, but as we searched published literatures from different sources regarding to HIV/AIDS and COVID-19 co-infection we have got few and those of which did not considered our combinations of interventions strategies in their model formulation. 

 The organization of the work may not be deemed necessary and can be removed. Section 2: mathematical model construction 

Reflection: We have removed it.

 Check the condition for modification parameter and what are their numerical values, seem to miss from Table 3. 

Reflection: We have incorporated All the modification parameters in Table 3 of the revised manuscript with reasonable values according to their definitions.

 In Figure 1: The schematic diagram of the COVID-19 and HIV Co-infection transmission - It’s not clear where do Cq come from (susceptible or infected class) - which mechanism is used to transfer individuals from class H_u to H_a at the rate of theta?. Are there any screening issues? Could you explain, please - How do individuals move from class S, H_p and C_v to class C_i how this possible? Is there any mechanism such as screening, testing etc used to validate this? - The class C_t is mentioned but does not appear in the model compartment - Class R is not defined. This model needs to be reworked 

Reflection: Individuals in the class C_q are coming from new born or immigrants abroad, individuals who are in the group H_u will transfer to the group H_a by educating the impact of screening mechanism to their health and living conditions, individuals who are in the class H_p are protected only from HIV infection and hence some individuals can be infected with COVID-19 that is why they transfer to the group C_i , individuals who are in the class C_v are vaccinated against COVID-19 infection but as we stated in the model assumption section due the COVID-19 serotype not covered by the COVID-19 vaccine some vaccinated individuals can be infected with COVID-19 and for each of these conditions there is a screening and testing mechanisms to check infected individuals previous non-infectious class. For classes C_i and R we have corrected our mistakes throughout the manuscript according to the reviewer comment.

 Analytical Result of the Models: This section requires revision to align with the model's formulation. 

Reflection: We have carried out according to the reviewer comments.

 Analysis of the Optimal Control Strategy: The manuscript appears to be lacking the following items, please consider incorporating them - Existence of an optimal control - Uniqueness of the Optimal Control Solution 

Reflection: In the revised manuscript we have incorporated the existence and uniqueness theorems.

 In Table 3. The numerical value for is 0.2 not 0,2 

Reflection: We have corrected accordingly.

 The figure captions lack adequate information and consistency in the heading of the figures 

Reflection: We have corrected accordingly.

 Writing language (typos, grammar etc.) and editing need to be seriously reviewed. 

Reflection: We have edited to correct our mistakes in the whole manuscript accordingly.

Review Report 2

 Abstract of the article should contain method, result and conclusion. In this article, there is no concept of optimal control theory. Please add about it.

Reflection: In the revised manuscript we have incorporated all the concepts mentioned by the reviewer. 

 In introduction part review some papers regarding optimal control of COVID-19, HIV and their confections using one paragraph since you are talking about optimal control as well in the paper.

Reflection: In the revised manuscript we have incorporated a review optimal control theory with one paragraph (sixth paragraph). 

 In equation (8), you concluded the LAS of DFE before express b interims of basic reproduction number. Therefore, express b interims of basic reproduction number before your conclusion.

Reflection: We strongly agree with the comment and we have expressed b in terms of the basic reproduction number in the revised manuscript.

 I know that GAS of EEP is somewhat difficult; however, you should add GAS of DFE of co-infections.

Reflection: As per the request of the reviewer using the well-known Castillo Chavez et al. literature given in reference number [61] we have shown the possibility of global asymptotic stability of the co-infection disease-free equilibrium point whenever its effective reproduction number is less than unity.

 In equation (14), what is the difference between forward and back-ward bifurcation? I need detail explanation.

Reflection: In the region where the model effective reproduction less than unity the model exhibit forward bifurcation means there is no stable endemic equilibrium point rather there is only stable disease-free equilibrium point, in this case the disease can be eradicated from the community in the near future. In other words in the same region the model exhibits backward bifurcation if and only if there is a stable positive endemic equilibrium point together with a stable disease-free equilibrium point in this case it is difficult to eradicate the disease from the community. 

 In optimal control theory, what does mean the set of admissible control functions? Why control variables are Lebesgue measurable? Why not Riemann measure? I need explanations.

 Questions on: The set of admissible Lebesgue measurable control functions is defined by

 Ω_u={(u_1 (t),u_2 (t),u_3 (t),u_4 (t))∈L^4:0≤u_1 (t),u_2 (t),u_3 (t),u_4 (t)≤1,t∈[0,T_f ]}. 

Admissible Control: A piecewise continuous control function u(t) which satisfies the control constraints on some time interval [T_0,T_f ] with range in the control region Ω_u is called admissible control.

Note: Our manuscript deals with real world situation and in real world situations resources are limited the controls have restriction in the limited resources called admissible control functions. 

Lebesgue measurable: Admissible control functions may be piecewise continuous functions which may have countable infinite number of discontinuities. In this case Riemann measure and Riemann integral cannot be applied since it only allows a finite number of discontinuities while Lebesgue measure and integral allows countable infinite number of discontinuities with the concept of almost everywhere (i.e., the set of points which do not satisfy the property has measure zero). That is why we defined the region Ω_u contains admissible Lebesgue measurable control functions.

 In theorem (8), the proof is not clear. Try to separate the proof and explain using Pontryagin’s minimum principle.

Reflection: We have revised it in section 5.2 of the revised manuscript accordingly

 In conclusion, there is no result which is connected with the concept of optimal control theory. Please add about it.

Reflection: We have revised the whole conclusion accordingly.

 What about the existence of optimal controls? State the theorem and leave the proof.

Reflection: In the revised manuscript we have stated the existence and uniqueness theorems with the proof uniqueness.

 In numerical simulation of optimal control part: describe the concept forward-backward fourth order Runge-Kutta method. I hope you used this numerical method. Moreover, in the objective functional of your problem, you minimized 4 populations, but your graph is only one compartment. Simulate all graphs side by side. See the articles mentioned below.

 Reflection: In the revised manuscript we have described the concept forward-backward fourth order Runge-Kutta method and incorporate additional simulations for single infections.

 Some work that can use for this literature especially on optimal control theory.

 T. D. Keno, O. D. Makinde and L. L. Obsu, Optimal Control and Cost Effectiveness Analysis of SIRS Malaria Disease Model with Temperature Variability Factor, Journal of Mathematical and Fundamental Sciences,53(1), 134-164, 2021.

 T. D. Keno, L. B. Dano, and O. D. Makinde. Modeling and Optimal Control Analysis for Malaria Transmission with Role of Climate Variability, Computaional and Mathematical Methods, 2022.

 T. D. Keno, O. D. Makinde and L. L. Obsu, Modelling and Optimal Control Analysis of Malaria Epidemic in the Presence of Temperature Variability, Asian-Europian Journal of Mathematics, 15(1), 2022.

Reflection: We have used all these relevant literatures suggested by the reviewers in our study and cited them in the revised manuscript.

---

## [Decision Letter · Decision Letter 1]

13 Mar 2023

PONE-D-23-01945R1Bifurcation and Optimal Control Analysis of HIV/AIDS and COVID -19

Co-Infection Model with Numerical SimulationPLOS ONE

Dear Dr. Kotola,

Thank you for submitting your manuscript to PLOS ONE. After careful consideration, we feel that it has merit but does not fully meet PLOS ONE’s publication criteria as it currently stands. Therefore, we invite you to submit a revised version of the manuscript that addresses the points raised during the review process.

We look forward to receiving your revised manuscript.

Kind regards,

Oluwole Daniel Makinde, PhD

Academic Editor

PLOS ONE

Journal Requirements:

Additional Editor Comments (if provided):

Minor revision required, see the reviewer's comment and revised accordingly.

Reviewers' comments:

Reviewer's Responses to Questions

**Comments to the Author**

1. If the authors have adequately addressed your comments raised in a previous round of review and you feel that this manuscript is now acceptable for publication, you may indicate that here to bypass the “Comments to the Author” section, enter your conflict of interest statement in the “Confidential to Editor” section, and submit your "Accept" recommendation.

Reviewer #1: All comments have been addressed

Reviewer #3: (No Response)

2. Is the manuscript technically sound, and do the data support the conclusions?

Reviewer #1: Yes

Reviewer #3: Yes

3. Has the statistical analysis been performed appropriately and rigorously? 

Reviewer #1: Yes

Reviewer #3: Yes

4. Have the authors made all data underlying the findings in their manuscript fully available?

Reviewer #1: Yes

Reviewer #3: Yes

5. Is the manuscript presented in an intelligible fashion and written in standard English?

Reviewer #1: Yes

Reviewer #3: Yes

6. Review Comments to the Author

Reviewer #1: The revised manuscript has improved significantly. The manuscript is well revised and written. Also, it was well organized.

Reviewer #3: -In the paper, a mathematical model for a transmission of HIV/AIDS and COVID-19 co-infection that incorporates protections and treatments for a population groups is formulated and analyzed. The results for non-negativity and boundedness of the co-infection model solutions are shown. The existence and stability of equilibriums using Routh-Hurwiz stability

criteria is also used. The Center Manifold criteria method are also used for analyzing the HIV/AIDS and COVID-19 co-infection model. It appears that the phenomenon of backward bifurcation characterizes the model on certain conditions.

Numerical simulations for both the deterministic model and the model incorporating optimal controls are done and show convergence for the model endemic equilibrium point whenever the model effective reproduction number is greater than unit.

-Theorem 6 states existence of the backward bifurcation but didn’t you simulate that process to show it visually?

-There are some typos in the text starting with Abstract with in the 3rd line: “is formulated and analyzed analyzed. “ and the paper needs a professional language editor to correct it in an improved english language.

-The main research topic in the manuscript together with general approach adopted and used by the authors deserve to be given credit.

-This is the revised version of the paper, and the mathematics look good, most of the theory is supported by proper simulations. The mathematical analysis is given and seems correctly performed. The paper may be considered after the above requests corrections.

7. PLOS authors have the option to publish the peer review history of their article (what does this mean?). If published, this will include your full peer review and any attached files.

Reviewer #1: **Yes: **Temesgen Duressa Keno (PhD)

Reviewer #3: **Yes: **EF Goufo

---

## [Author Response · Author response to Decision Letter 1]

27 Mar 2023

Reviewer comments and our reflections for the comments

First of all we would like to say thank you so much for taking the time to review our manuscript and provide us with your valuable feedback. We genuinely appreciate your thoughtful contributions and values the effort you have put into making this project a success. Your feedback and insight have been invaluable, and we are truly grateful for your support and input. Thank you for believing in us and helping us reach our goals. We have stated the requested correction and our reflection as follow.

Reviewer #3: 

1. Theorem 6 states existence of the backward bifurcation but didn’t you simulate that process to show it visually?

Reflection: As per the request of the reviewer comment in the revised manuscript we have demonstrated the existence of back ward bifurcation using simulation.

2. There are some typos in the text starting with Abstract with in the 3rd line: “is formulated and analyzed analyzed. “and the paper needs a professional language editor to correct it in an improved english language.

Reflection: We have edited to correct our mistakes in the whole manuscript accordingly.

3. The main research topic in the manuscript together with general approach adopted and used by the authors deserve to be given credit.

Reflection: We are strongly believe that all the authors guide us deserve credit so that, in the revised manuscript we have incorporated all the concepts mentioned.

---

## [Decision Letter · Decision Letter 2]

10 Apr 2023

Bifurcation and Optimal Control Analysis of HIV/AIDS and COVID -19

Co-Infection Model with Numerical Simulation

PONE-D-23-01945R2

Dear Dr. Kotola,

We’re pleased to inform you that your manuscript has been judged scientifically suitable for publication and will be formally accepted for publication once it meets all outstanding technical requirements.

Kind regards,

Oluwole Daniel Makinde, PhD

Academic Editor

PLOS ONE

Additional Editor Comments (optional):

Reviewers' comments:

Reviewer's Responses to Questions

**Comments to the Author**

1. If the authors have adequately addressed your comments raised in a previous round of review and you feel that this manuscript is now acceptable for publication, you may indicate that here to bypass the “Comments to the Author” section, enter your conflict of interest statement in the “Confidential to Editor” section, and submit your "Accept" recommendation.

Reviewer #3: All comments have been addressed

2. Is the manuscript technically sound, and do the data support the conclusions?

Reviewer #3: Yes

3. Has the statistical analysis been performed appropriately and rigorously? 

Reviewer #3: Yes

4. Have the authors made all data underlying the findings in their manuscript fully available?

Reviewer #3: Yes

5. Is the manuscript presented in an intelligible fashion and written in standard English?

Reviewer #3: Yes

6. Review Comments to the Author

Reviewer #3: The authors managed to improve the quality of the paper according to previous comments and the paper is ready to be put on the public domain.

7. PLOS authors have the option to publish the peer review history of their article (what does this mean?). If published, this will include your full peer review and any attached files.

Reviewer #3: No
